# Analysis of the Influence of Thermal Loading on the Behaviour of the Earth's Crust

**Ivo Wandrol** [1,*] **, Karel Frydrýšek** [2,3] **and Daniel Čepica** [2,3]

1   Institute of Physics in Opava, Silesian University in Opava, Bezručovo Náměstí 1150/13, 746 01 Opava, Czech Republic

2   Department of Applied Mechanics, Faculty of Mechanical Engineering, VSB—Technical University of Ostrava, 17. Listopadu 2172/15, 708 00 Ostrava, Czech Republic; karel.frydrysek@vsb.cz (K.F.); daniel.cepica@vsb.cz (D.Č.)

3   Institute of Emergency Medicine, Faculty of Medicine, University of Ostrava, Syllabova 19, 703 00 Ostrava-Vítkovice, Czech Republic

*   Correspondence: ivo.wandrol@slu.cz

**Abstract:** The article focuses on the deformation and strain-stress analysis of the Earth's crust under external thermal loading. More specifically, the influence of cyclic changes in the surface temperature field on the stress and displacement inside the crust over a two-year time span is investigated. The finite element program MSC.Marc Mentat was used to calculate the stresses and displacements. For practical analysis reasons, the Earth's crust is simplified as a planar, piecewise homogeneous, isotropic model (plane strain), and time-varying temperature functions of illumination (thermal radiation) from the Sun are considered in the local isotropy sections of the model. Interaction between the Earth's crust and mantle is defined by the Winkler elastic foundation. By applying a probabilistic approach (Monte Carlo Method), a new stochastic model of displacements and stresses and new information on crustal displacements relative to the Earth's mantle were obtained. The results proved the heating influence of the Sun on the Earth's crust and plate tectonics.

**Keywords:** geomechanics; Earth's crust; Finite Element Method; Sun heating (radiation); stress; displacement; elastic foundation; stochastic approach; Earth's crust; tectonics

## 1. Introduction

The geological structure of planet Earth is complicated and full of 'unknowns'. Therefore, it is suitable to make a discretization for numerical modelling; see Figure 1 [1–3].

The Earth's crust is affected by many external influences, such as tidal forces, cyclic changes in crustal surface temperatures caused by the Sun, recurrent changes in atmospheric air pressure, the transmission of ocean and sea water mass waves to the Earth's crust, and geological processes within the core and mantle of the planet [2–9].

This paper focuses on the effect of cyclic surface temperature changes on the stress and strain of the Earth's crust. The crust (comprising lithospheric plates) is periodically heated (thermally deformed and stressed) by radiation induced by stellar physical processes inside the Sun. These are mainly diurnal periods (alternation of day and night—24 h cycle) or annual periods (alternation of seasons in about 365.4 days). Both of these periods have been applied to the input of the surface temperature of the Earth [9–12].

Due to the complexity of the whole process, the Finite Element Method (FEM) is used and the problem is solved as a planar one [2–5]. The Winkler elastic foundation model, which is commonly used in mechanics, is used for the interaction of the Earth's crust and mantle, as shown in Figure 1 [9,13–17]. However, application of the Winkler elastic foundation in geomechanics is a new way of solving the problem.

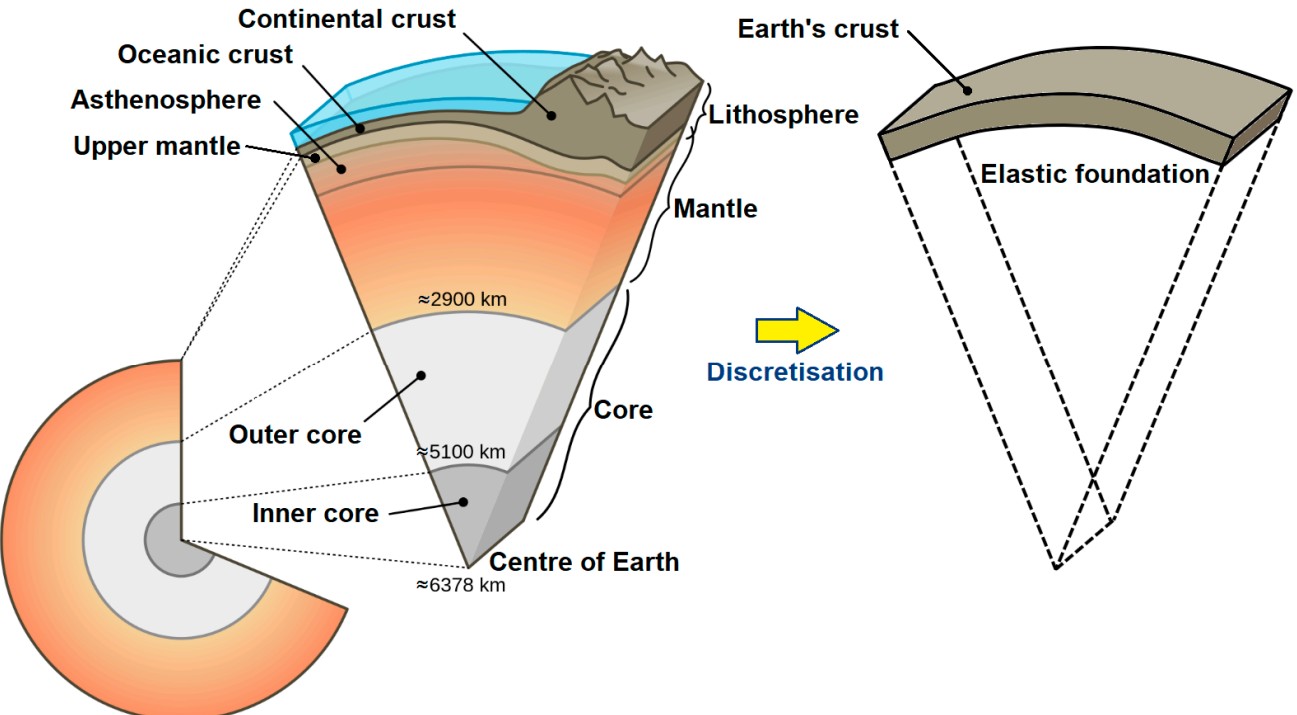

**Figure 1.** Structure of the Earth and the possible discretization applied (not to scale).

The Earth's crust is an anisotropic, heterogeneous upper part of the geoid and is simplified in the analysis as a planar, piecewise, homogeneous, isotropic model with time-varying temperature functions of illumination, i.e., thermal radiation, from the Sun. The latter are considered in the local isotropy sections of the model.

The generally anisotropic crust is partitioned into piecewise isotropic regions and the whole problem is solved as a 2D (plane strain) problem. The theory of large deformations, i.e., true (logarithmic) strains and true stresses, was used because it gives more accurate results, but leads to the solution of a non-linear problem. This is a non-stationary coupled (thermal + structural) problem and more details are given in [9,13–15].

A new and modern probabilistic (stochastic) approach is applied, which respects the real variability of the obtained records of thermal radiation from the Sun.

There are not enough global models of the Earth's crust, in connection with the FEM and stochastic approaches. This article fills the information gap. However, 2D models are commonly used in geomechanics [9,12].

In the first step, the influence of gravitational and tidal effects from the Sun and Moon, and the effect of rotational and centrifugal forces on the Earth's crust, are neglected in the model. The reason for this is the thermal heating of the crust, which is the focus and research of this article. We focused on thermal heating of the crust because it is usually neglected or not considered, but it has a significant influence. Nevertheless, a somewhat simple probabilistic model has been developed which respects the real variability of input and output quantities [9,15–25].

Other possible approaches connected with groundwater investigations and uncertainty quantification are presented in [12,23]. In reference [23], there is a state-of-the-art in approach for uncertainty quantification in geomechanics. However, this approach cannot yet be applied to the complicated problem of crust-mantle interaction because of the lack of global information about the crust.

It has recently been proved that the Sun's heating of the Earth's surface can have a partial and important influence on the creation and propagation of plate tectonics.

So far, there have been no simpler nonlinear geomechanical stochastic/probabilistic models incorporating crust-mantle interaction, with a focus on deformation and stress

states in the Earth's crust, induced by solar radiation. This is the main contribution of this paper.

Our work builds on simpler, previously developed, models [9,18,19,26–31].

Alternatively, the findings or procedures presented in [32], which were carried out for an area in Sweden, can also be used, but their application to the entire surface of the Earth is too challenging and complicated.

The nomenclature for all of the variables and abbreviations used is presented at the end of this article.

## 2. Materials and Methods

If the Earth is to be considered as a sphere in the calculations, the solution will be challenging, so it is appropriate to introduce a simpler planar model in the initial approximation that can be extended in the future.

However, if the dimensions of the heat source (the Sun) and its distance from the Earth are taken into account, it can be concluded that the heat rays incident on the Earth are almost parallel. This approach is common in mechanics.

An acceptable simplification for the calculations can be achieved by replacing the Earth's crust by an infinite hollow cylinder (see Figure 2), the so-called 'Earthcylinder', and defining the heating or cooling of the Earth's outer surface as a time-varying heat function $T = f(\varphi, t)$, where $\varphi$ is the angle /deg/ and $t$ is time /s/ [9].

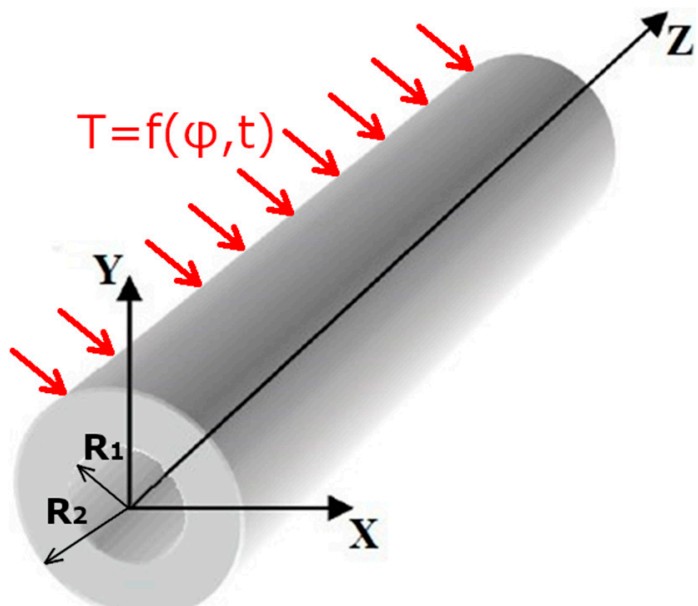

**Figure 2.** Model of the Earth as an infinite hollow cylinder, the so-called 'Earthcylinder' with heating from the Sun (not to scale).

The Earth's crust is composed of lithospheric plates and is mainly made up of rocks and minerals, where various faults, fractures, and other geological formations occur. In addition to rocks and minerals, there are also gases and water. It is, therefore, an inherently highly anisotropic, heterogeneous and inhomogeneous material [2,32–39].

The creation of such a material model would be very complicated, and so some appropriate simplifications are adopted in the calculations. From the point of view of material properties, our computational model is a composite [4,5,9]. The model of the Earth's crust is divided into 24 sections with different material properties; it is a piecewise isotropic homogeneous material model, which appears to be anisotropic from the outside. On the surface of each material section, there are temperature functions $T_1, T_2 \ldots, T_{24} = f(\varphi, t)$, which correspond to time-varying temperature values over a two-year period, according to [6,7], as shown in Figure 3.

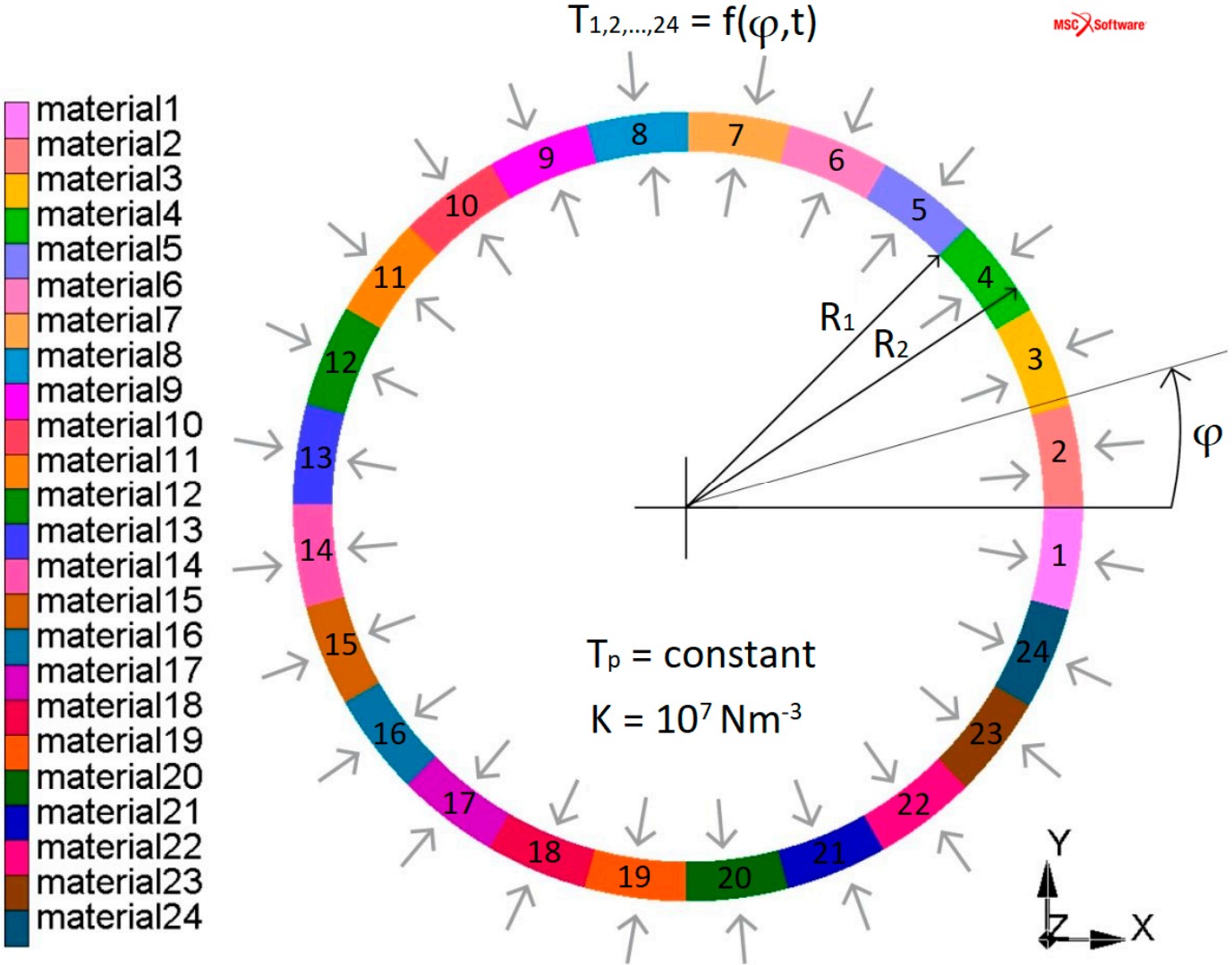

**Figure 3.** 'Earthcylinder' with materials and temperature loads (not to scale).

For the FEM calculation, the problem is treated as a planar problem with an assumed plane strain. In Figures 3 and 4, the crustal thickness is highly enlarged (for clarity), and the planar model is shown as being divided into 24 sections, with assigned material properties and loading indicated from the temperature. The angular perimeter is divided by 15 degrees, with a different material in each of the 15 degrees, as shown in Figures 3 and 4.

The 'Earthcylinder' is a model of a cylinder replacing the earth's crust with constant inner and outer radii, $R_1 = 6348$ km and $R_2 = 6378$ km, as shown in Figures 1–3.

Below the crust, i.e., below the radius $R_1$, is the upper mantle, whose assumed constant temperature is $T_p$. This temperature is mostly transferred to the inner radius of the crust by thermal conduction, as shown in Figures 3 and 4.

In the model, the upper mantle is replaced by a Winkler elastic foundation with the Modulus of the Foundation $K = 10^7$ Nm$^{-3}$, as shown in Figures 1 and 3 [9,13–15,19]. The elastic foundation is also applied in many engineering problems [24–27].

The outer radius of the model $R_2$ is affected by the temperature $T = f(\varphi, t)$. This temperature depends on the angle $\varphi$ and the time t. Such a dependence respects the fact that the temperature varies at different locations on the Earth's crust at different times, i.e., there are temperature differences between day and night or, possibly, seasons [9].

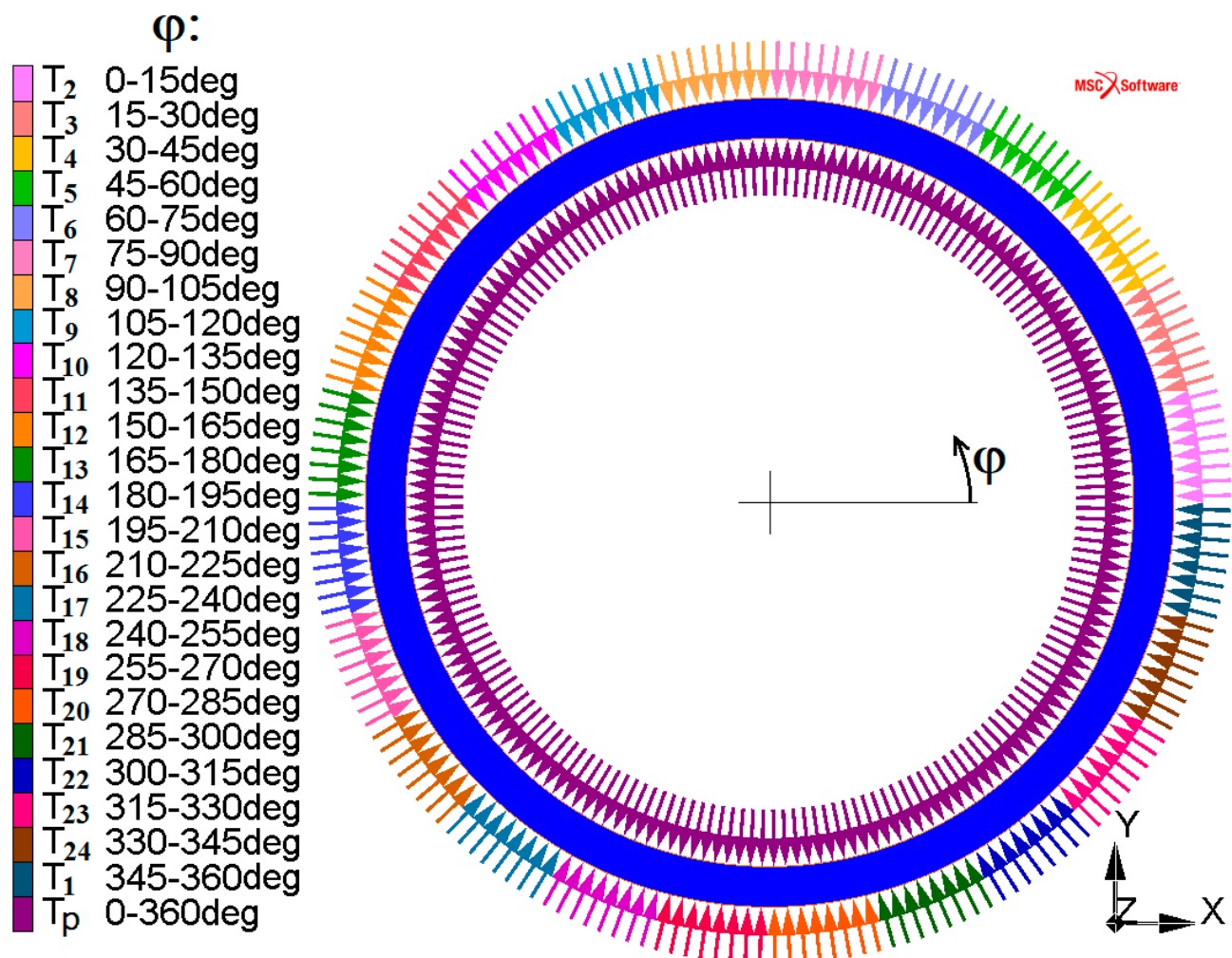

**Figure 4.** 'Earthcylinder' with temperature loads (not to scale).

The material properties of individual sections of the Earth's crust, as shown in Figure 3, vary within given ranges and are determined by random combinations:

- Young's modulus E $\in \langle 3 \times 10^{10}; 4.5 \times 10^{10} \rangle$ Pa;
- Poisson number $\mu \in \langle 0.30; 0.35 \rangle$;
- Density (mean value) $\rho$ = 2760 kgm$^{-3}$;
- Conductivity (thermal conductivity coefficient) $\lambda$ = 3 Wm$^{-1}$K$^{-1}$;
- Specific heat c = 1100 Jkg$^{-1}$K$^{-1}$;
- Thermal expansion coefficient $\alpha \in \langle 3.5 \times 10^{-5}; 4.5 \times 10^{-5} \rangle$ K$^{-1}$.

For the individual sections 1–24 in Figure 3, the material properties are determined according to the ranges mentioned above. The material properties are given in Table 1. The real material properties of the materials contained in the earth's crust are given in [9,37–41].

### 2.1. Temperature Loading

A constant (time independent) mean temperature $T_p$ = 648 K is considered to be on the inner surface of the cylinder, i.e., the Earthcylinder, at a depth of 30 km in the Earth's crust, see [9,41].

**Table 1.** Material properties of individual sections of the planar crustal model of the Earth's crust.

| Section | Angle φ /deg/ | Young's Modulus E/Pa/ | Poisson Number /1/ | Density /kg·m$^{-3}$/ | Conductivity /W·m$^{-1}$·K$^{-1}$/ | Specific Heat /J·K$^{-1}$·kg$^{-1}$/ | Thermal Expansion /K$^{-1}$/ |
|---|---|---|---|---|---|---|---|
| 1 | 345–360 | $3.00 \times 10^{10}$ | 0.30 | 2760 | 3 | 1100 | $3.50 \times 10^{-5}$ |
| 2 | 0–15 | $3.50 \times 10^{10}$ | 0.35 | 2760 | 3 | 1100 | $4.00 \times 10^{-5}$ |
| 3 | 15–30 | $4.00 \times 10^{10}$ | 0.30 | 2760 | 3 | 1100 | $4.50 \times 10^{-5}$ |
| 4 | 30–45 | $4.50 \times 10^{10}$ | 0.35 | 2760 | 3 | 1100 | $3.50 \times 10^{-5}$ |
| 5 | 45–60 | $3.00 \times 10^{10}$ | 0.30 | 2760 | 3 | 1100 | $4.00 \times 10^{-5}$ |
| 6 | 60–75 | $3.50 \times 10^{10}$ | 0.35 | 2760 | 3 | 1100 | $4.50 \times 10^{-5}$ |
| 7 | 75–90 | $4.00 \times 10^{10}$ | 0.30 | 2760 | 3 | 1100 | $3.50 \times 10^{-5}$ |
| 8 | 90–105 | $4.50 \times 10^{10}$ | 0.35 | 2760 | 3 | 1100 | $4.00 \times 10^{-5}$ |
| 9 | 105–120 | $3.00 \times 10^{10}$ | 0.30 | 2760 | 3 | 1100 | $4.50 \times 10^{-5}$ |
| 10 | 120–135 | $3.50 \times 10^{10}$ | 0.35 | 2760 | 3 | 1100 | $3.50 \times 10^{-5}$ |
| 11 | 135–150 | $4.00 \times 10^{10}$ | 0.30 | 2760 | 3 | 1100 | $4.00 \times 10^{-5}$ |
| 12 | 150–165 | $4.50 \times 10^{10}$ | 0.35 | 2760 | 3 | 1100 | $4.50 \times 10^{-5}$ |
| 13 | 165–180 | $3.00 \times 10^{10}$ | 0.30 | 2760 | 3 | 1100 | $3.50 \times 10^{-5}$ |
| 14 | 180–195 | $3.50 \times 10^{10}$ | 0.35 | 2760 | 3 | 1100 | $4.00 \times 10^{-5}$ |
| 15 | 195–210 | $4.00 \times 10^{10}$ | 0.30 | 2760 | 3 | 1100 | $4.50 \times 10^{-5}$ |
| 16 | 210–225 | $4.50 \times 10^{10}$ | 0.35 | 2760 | 3 | 1100 | $3.50 \times 10^{-5}$ |
| 17 | 225–240 | $3.00 \times 10^{10}$ | 0.30 | 2760 | 3 | 1100 | $4.00 \times 10^{-5}$ |
| 18 | 240–255 | $3.50 \times 10^{10}$ | 0.35 | 2760 | 3 | 1100 | $4.50 \times 10^{-5}$ |
| 19 | 255–270 | $4.00 \times 10^{10}$ | 0.30 | 2760 | 3 | 1100 | $3.50 \times 10^{-5}$ |
| 20 | 270–285 | $4.50 \times 10^{10}$ | 0.35 | 2760 | 3 | 1100 | $4.00 \times 10^{-5}$ |
| 21 | 285–300 | $3.00 \times 10^{10}$ | 0.30 | 2760 | 3 | 1100 | $4.50 \times 10^{-5}$ |
| 22 | 300–315 | $3.50 \times 10^{10}$ | 0.35 | 2760 | 3 | 1100 | $3.50 \times 10^{-5}$ |
| 23 | 315–330 | $4.00 \times 10^{10}$ | 0.30 | 2760 | 3 | 1100 | $4.00 \times 10^{-5}$ |
| 24 | 330–345 | $4.50 \times 10^{10}$ | 0.35 | 2760 | 3 | 1100 | $4.50 \times 10^{-5}$ |

### 2.1.1. Initial Conditions

The initial preheating condition of the model determines the temperature at the beginning of the analysis in MSC.Marc Mentat 2006 software. On the inner surface, the temperature $T_p$ is prescribed, and on the outer surface of the crust, the temperature $T_c = 287$ K is prescribed. From the mentioned initial temperatures $T_p$ and $T_c$, the initial temperature distribution $T_{initial} \in \langle T_c; T_p \rangle$ is acquired. The initial temperature is independent of the angle φ but dependent on the radius R, as shown in Figure 5.

### 2.1.2. Boundary Conditions

The temperature boundary condition $T = f(\varphi, t)$ (i.e., $T_1, T_2 \ldots, T_{24} = f(\varphi, t)$, as shown in Figures 2–4), depends on time t and the angular dimension of the Earth and acts on the outer surface of the crust. This boundary condition simulates the cyclic temperature changes based on the daily cycle of the Earth's rotation and the annual cycle of the planet's orbit around the Sun. An example of $T_2$ dependence loading for 2 years and 5 days is shown in Figure 6.

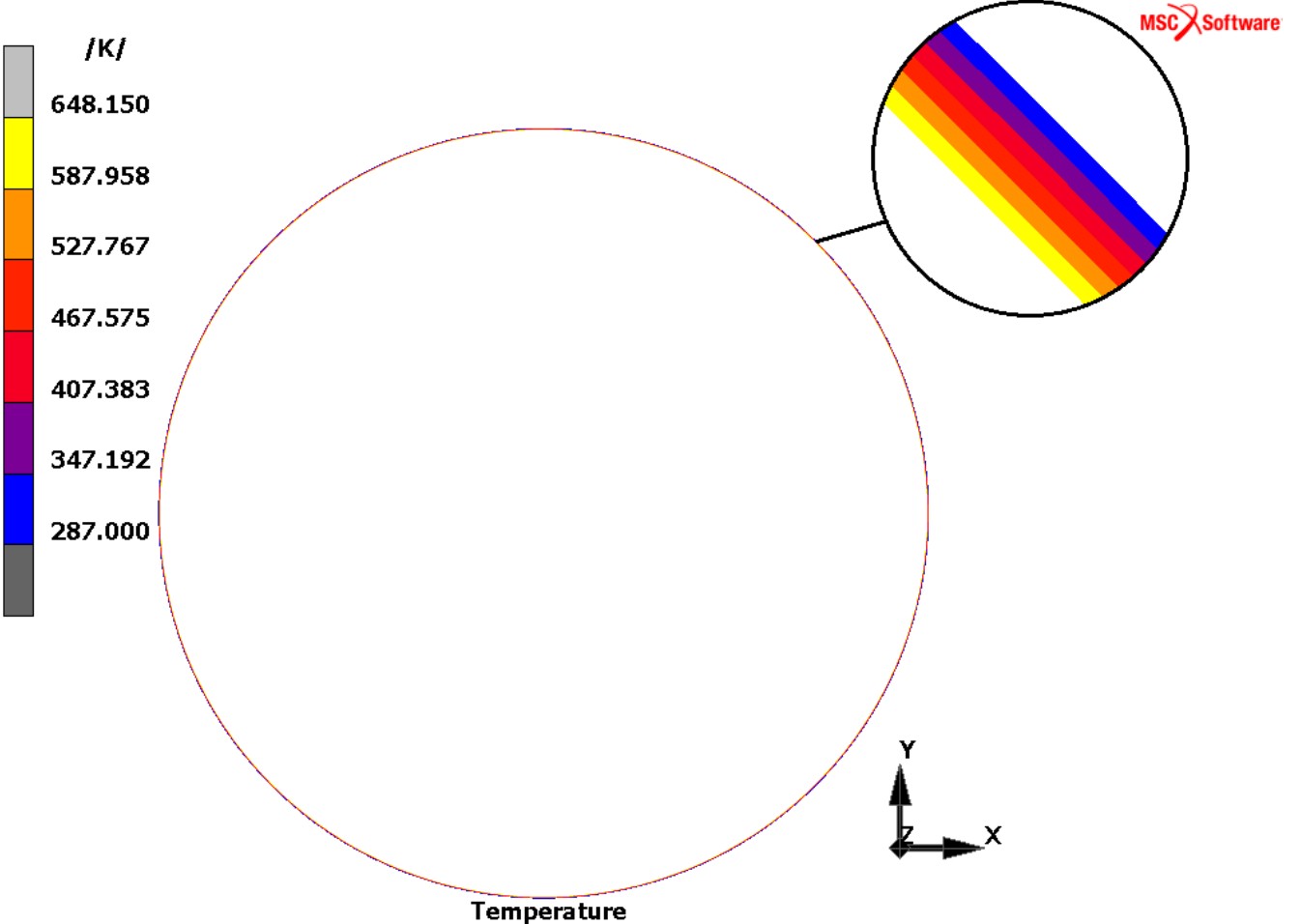

**Figure 5.** Earthcylinder and its initial temperature distribution $T_{initial}$.

For the calculation, surface temperatures from the ALA monitoring system [10,11] (in addition to surface temperature and air temperature) were taken from stations located between latitudes of approximately 40 and 50 degrees north for the period October 2007 to August 2009, at 15 min increments.

The basic characteristic values of the statistical data of temperature measurements $T_1, T_2 \ldots, T_{24} = f(\varphi, t)$ are:

- Minimum temperature = 263.65 K,
- Mean temperature = 282.20 K,
- Median temperature = 282.15 K,
- Standard deviation of temperature = 7.68 K,
- Maximum temperature = 303.35 K.

The data were then processed for importing into the MSC.Marc Mentat 2006 software [42]. The resulting time-dependent temperature series contained 35,424 values, with 110 real temperature anomalies where the temperature continuity check (large variability between two measurements) was not met. These were mainly large fluctuations in air temperature, which could have been caused by the outbreak of a severe storm, etc. The surface temperature followed the air temperature pattern. In this case, the above-mentioned anomalies occurred only in the spring and summer periods, from 3 April 2008 to 30 August 2008. Of course, these anomalies, which are real in nature, were also used in the solution.

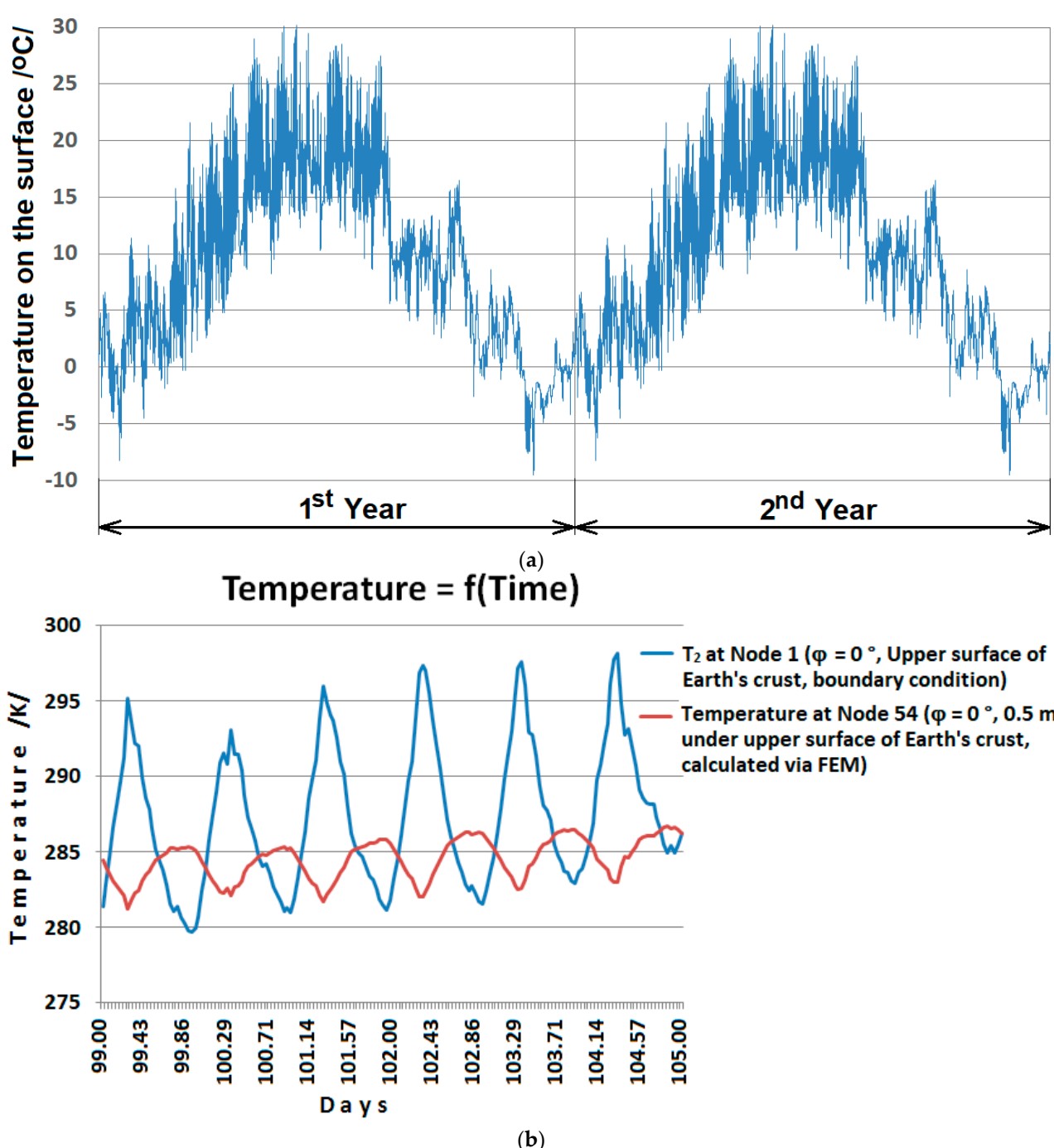

**Figure 6.** Temperature $T_2$ distribution (**a**) Boundary condition—for whole 2 years at node 1; (**b**) Boundary condition at node 1 and calculated temperature at node 54 for 99th to 105th day of solution.

It can, therefore, be concluded that the temperature variability is significantly higher in the rising temperature phase than in the falling temperature phase. A similar phenomenon could be observed in the diurnal time cycle. For this reason, the idea of modelling the temperature function using goniometric functions was abandoned as it would introduce errors (the application of Fourier series will not bring accuracy and is not necessary).

The processed data were used for the section $\varphi \in \langle 345°; 360° \rangle$. For all subsequent sections, the entire time series was successively shifted by one hour. For Section 2 of Figure 3 ($\varphi \in \langle 0°; 15° \rangle$), the date 1 February 2008 0:00:00 corresponded to the original date

of 1 February 2008 01:00:00 and, for each subsequent section, the time series is shifted by another hour, as already mentioned.

The different temperature boundary conditions in each section represent a virtual rotation of the model, i.e., the change in temperature simulates the rotation of the Earth around the Sun.

The temperature time series, however, had to be reduced for importing into the MSC.Marc Mentat software due to the large amount of data, so that the original time step of 15 min had to be increased to a time step of 3519 s (i.e., 58 min 39 s). This leads to the faster solution by FEM.

Since the initial surface temperature of the model is constant around the perimeter and the boundary temperature conditions are angle dependent, it is necessary to let the model settle at the beginning of the analysis. According to the results obtained, the model is considered to be steady after three days of loading, i.e., the values from 4 February 2008 00:48:37 are taken for processing the results. More information regarding the boundary conditions is given in [9].

## 2.2. Finite Element Mesh

Figure 7 shows the finite element mesh used for the calculation. At the transition points between the two sections, i.e., different material properties and surface temperatures, bias subdivision is used with refinement of the mesh towards the outer surface, since larger temperature variations are expected near the outer surface than at greater depths. The FE mesh contains 8973 quadratic elements and 10,116 nodes.

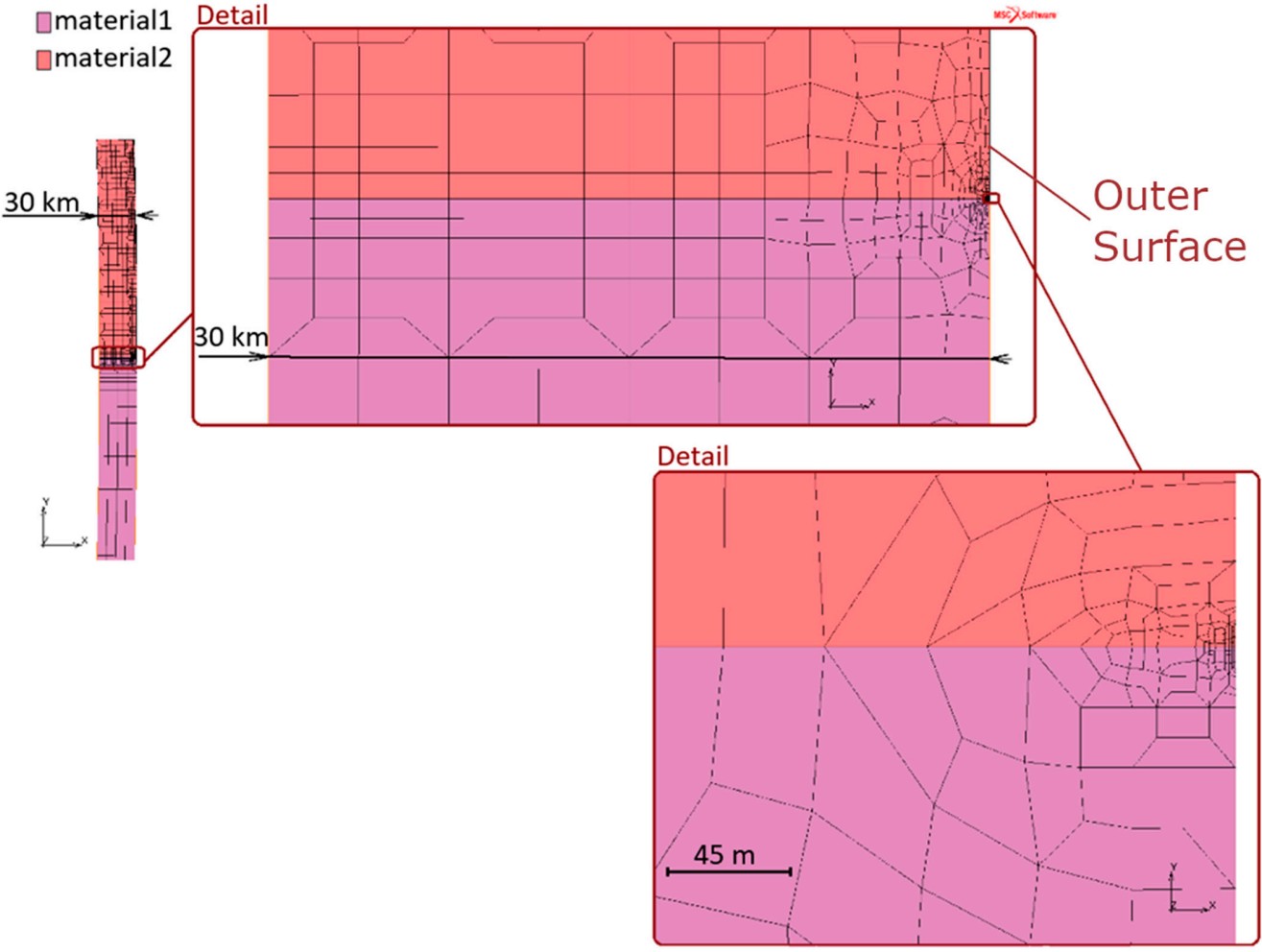

**Figure 7.** Detail of biased mesh (refinement towards the surface of the Earth).

### 3. Results

From the FE calculation performed on an Earthcylinder, the mechanical stress and displacement distributions were evaluated. The equivalent von Mises mechanical stress (HMH) $\sigma_{HMH}$ /Pa/ is calculated according to Equation (1)

$$\sigma_{HMH} = \sqrt{\sigma_1^2 + \sigma_2^2 + \sigma_3^2 - (\sigma_1\sigma_2 + \sigma_2\sigma_3 + \sigma_1\sigma_3)}, \tag{1}$$

where $\sigma_{1,2,3}$ /Pa/ are the principal stresses, see [43].

The total displacement $u$ /m/ can be expressed by

$$u = \sqrt{u_x^2 + u_y^2}, \tag{2}$$

where $u_x$ /m/ is the displacement in the X-axis direction, and $u_y$ /m/ is the displacement in the Y-axis direction, see Figure 2 or Figure 3.

Equations (1) and (2) were applied in the finite element analysis (FEA).

#### 3.1. Results from FEA

As for the finite element model, the accuracy was checked by comparison. First, a coarser finite element mesh with a longer time step was solved. Then a finer mesh with a shorter time step was solved, e.g., this finite element id described in Figure 7. There was no significant difference between these solutions and the presented solution can then be declared sufficiently accurate. Because of problems where the analytical solution is not known, there is no other way to check the accuracy, i.e., for our task, the exact solution suitable for comparison with our numerical results cannot be obtain.

#### 3.1.1. Stress Analysis

Figure 8 shows the depth evolution of the equivalent stress $\sigma_{HMH}$ at the interface of sections 1 and 2 of Figure 3 (i.e., $\varphi = 0°$). In Figure 8, the stress attenuation towards depth is evident, as was the case for the other sections. The trend of decreasing $\sigma_{HMH}$ towards the Earth's interior can also be partly explained by the all-round pressure stress states that characterize matter deep underground or in water. The resulting $\sigma_{HMH}$ is given by Equation (1).

Figure 9 shows the time history of the equivalent stress $\sigma_{HMH}$ at a depth of 2812.5 m between sections 1 and 2 of Figure 3 ($\varphi = 0°$).

From a detailed analysis of the results, it was found that the variance of stress is significantly larger than the variance of temperature over the annual cycle of temperature changes.

In the following text, the stresses are no longer evaluated. The evaluation carried out in [9] is not the main focus of this article.

#### 3.1.2. Displacement

Figure 10 shows the depth profile of the total displacement between sections 1 and 2 of Figure 3 ($\varphi = 0°$), where the influence of the annual cycle of temperature change is clearly evident.

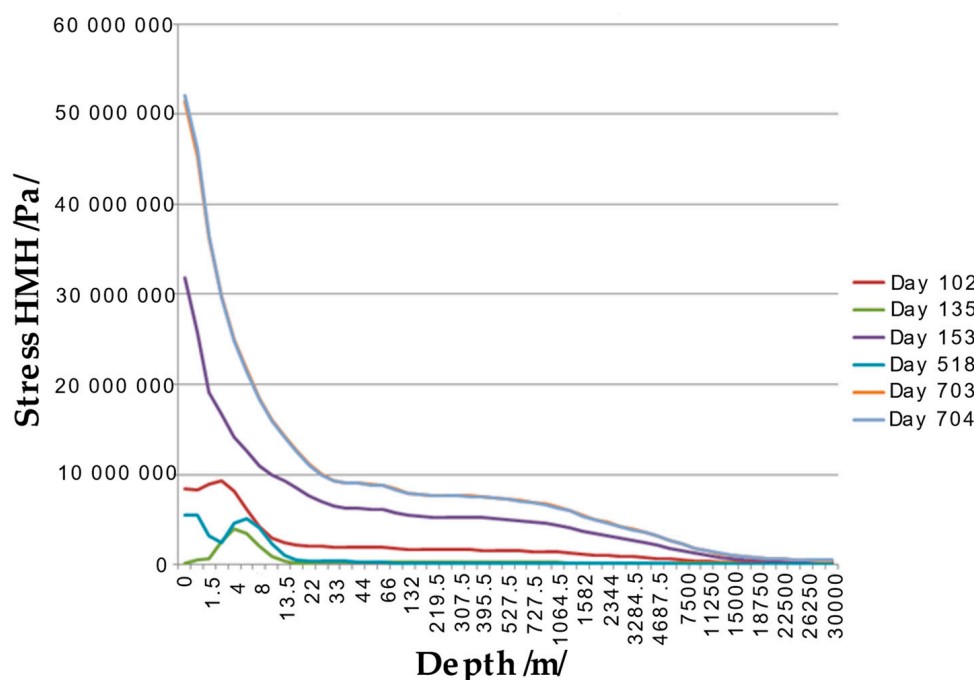

**Figure 8.** Plot of depth evolution of the equivalent von Mises stress (HMH) /Pa/ at the interface of sections 1 and 2 of Figure 3 (φ = 0°).

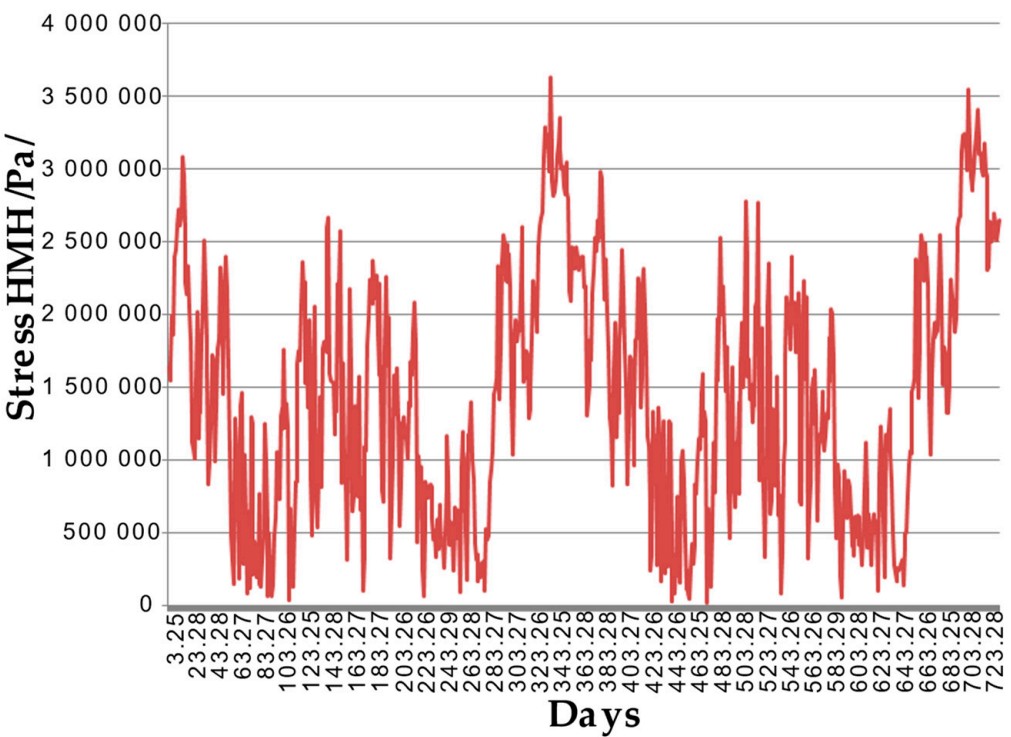

**Figure 9.** Plot of time dependence of the von Mises equivalent stress (HMH) /Pa/ at a depth of 2812.5 m between sections 1 and 2 of Figure 3 (φ = 0°).

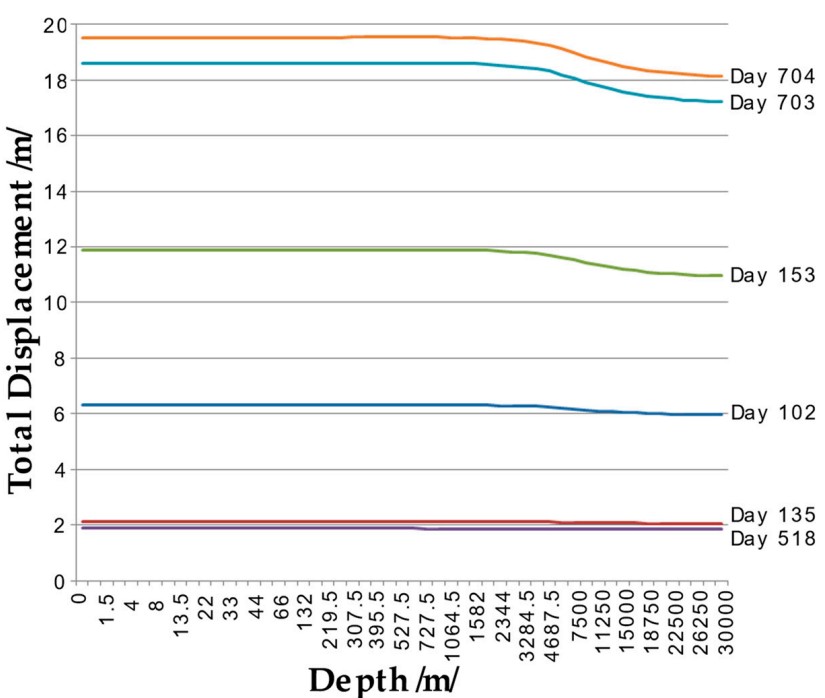

**Figure 10.** Depth profile of total displacement /mm/ between sections 1 and 2 of Figure 3 ($\varphi = 0°$).

Over the course of a year, the total displacement of the Earth's crust changes substantially by tens of meters. These changes are imperceptible to the naked eye, except in the case of earthquakes or tectonic plate ruptures, when they are almost instantaneous and visible.

Figure 11 shows the time dependence of the total displacement at a depth of 2812.5 m between sections 1 and 2 of Figure 3 ($\varphi = 0°$). A large variance in displacement can be observed here too, i.e., the influence of the annual cycle of temperature changes.

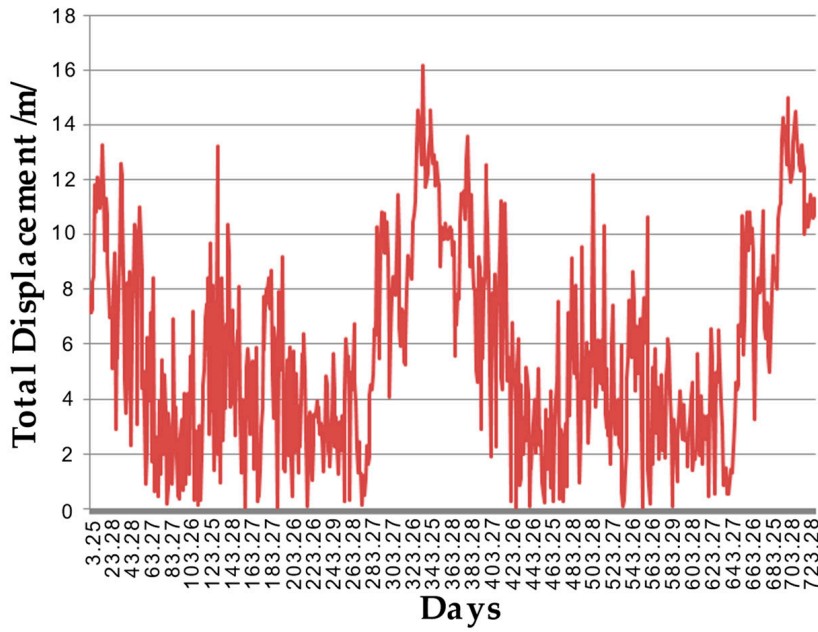

**Figure 11.** Plot of time dependence of total displacement /mm/ at a depth of 2812.5 m between sections 1 and 2 of Figure 3 ($\varphi = 0°$).

The time records of total stresses and displacements, as shown in Figures 9 and 11, are similar in nature.

Figure 12 shows the vectors of total displacement at the outer surface nodes at the interfaces of each section and, additionally, at the nodes in the middle of each section at time 28,813,890 s (i.e., 333.494 days). The resulting displacement is given by Equation (2).

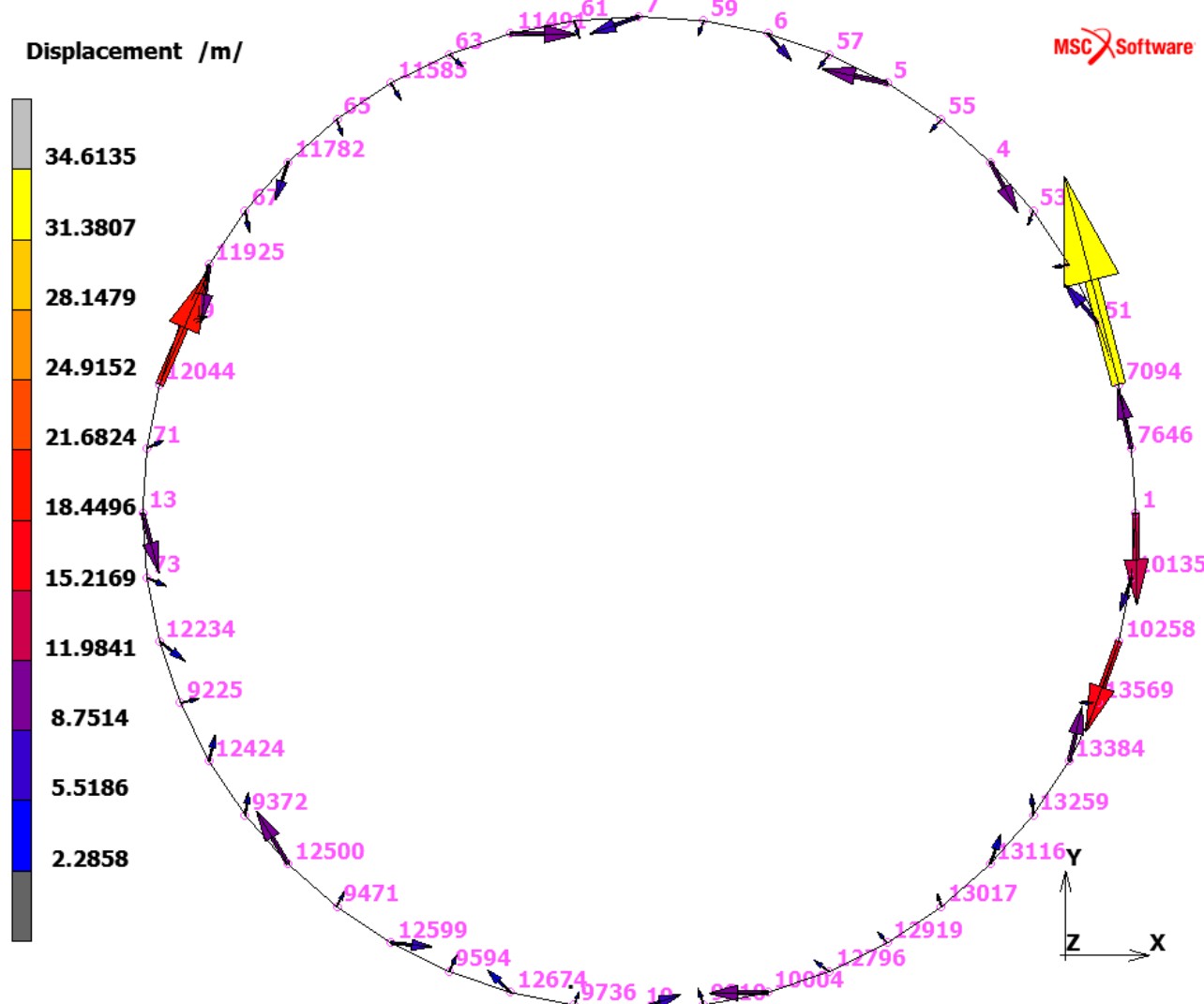

**Figure 12.** Vectors of the resulting displacement at the chosen outer surface nodes at a time of 28,813,890 s (i.e., 333.494 days), i.e., on 30 December 2008 with the numerical designation of the surface nodes.

In several nodes, the predominant influence of the tangential component of the displacement over the radial component is noticeable; this phenomenon can be observed most clearly at node '7094' (the boundary of sections 2 and 3 of Figure 3, $\varphi = 15^\circ$). The dominance of tangential drift at these nodes may be due to a change in material properties or a manifestation of possible continental drift.

For a stochastic evaluation, the displacement is divided into the aforementioned tangential and radial components.

Figure 12 shows the considerable variability and, to some extent, inhomogeneity in crustal displacement, in terms of direction and magnitude. On other days, this variability was also evident but with different magnitudes and directions. This variability is typical in the Earth's crust because there are clearly distinct deformations, e.g., mountains, mountain ranges, trenches, plate breaks, etc.

It is appropriate to statistically evaluate the mentioned experience and draw conclusions.

### 3.2. Stochastic Evaluation of Results

For clearer outputs of the FEA, the results are processed in the form of statistical histograms. Anthill software [44,45] is used to plot the histograms, which give the total displacements at each node over the whole analyzed time. Thus, sufficiently accurate statistical records of the data are obtained. Statistical processing of the stresses was carried out in [9].

All Monte Carlo calculations were performed for $2 \times 10^6$ pseudo-random simulations.

Tangential and Radial Displacement Components

Figures 13–15 show examples of histograms and distribution functions of radial and tangential displacement components for nodes '1', '7094', and '12,044' on the outer surface (labelled according to Figure 11).

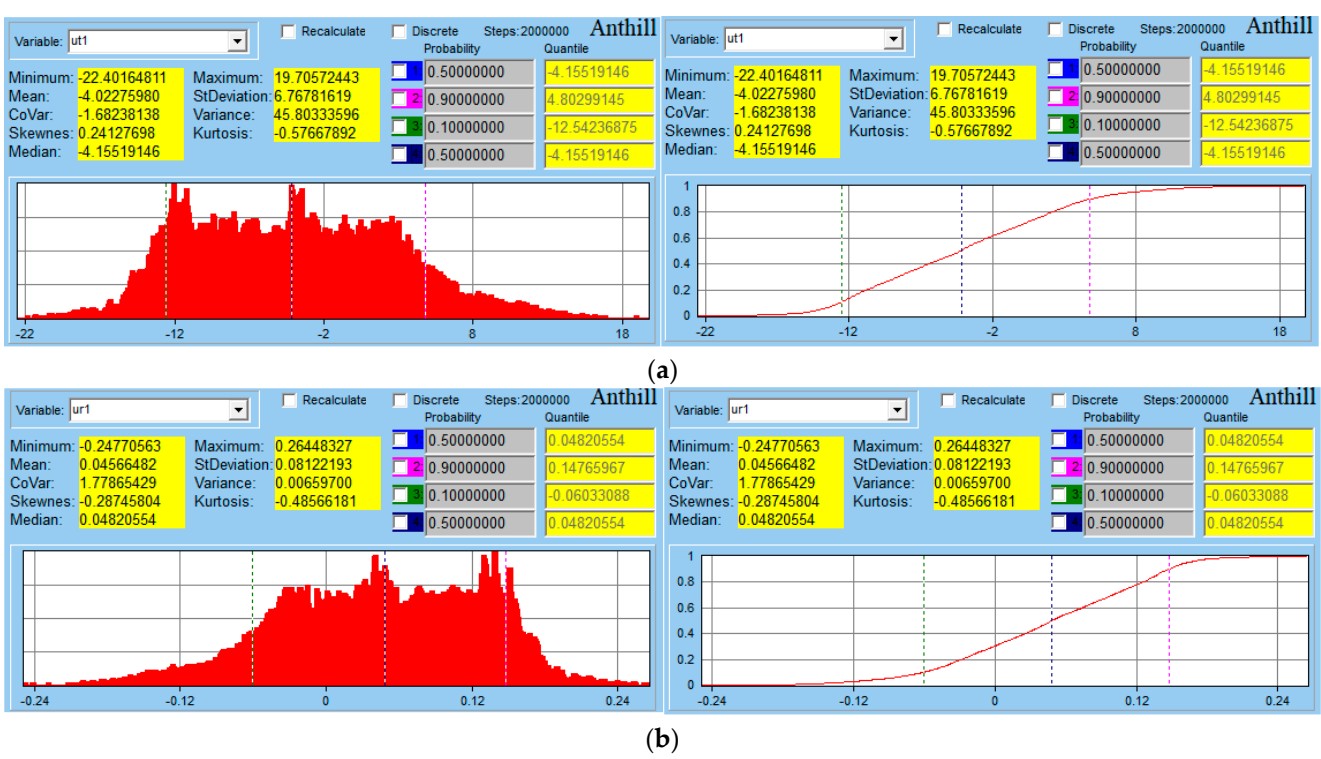

(**a**)

(**b**)

**Figure 13.** Histograms and distribution functions for node '1' (**a**) tangential components of crustal surface displacement, (**b**) radial components of crustal surface displacement (sw MSC.Marc Mentat and Anthill).

Tables 2 and 3 show a statistical evaluation of the tangential and radial displacements of the Earth's crustal surface.

Figure 16 shows the tangential schematic displacement histograms for all nodes on the outer surface according to Figure 11, i.e., with angular divisions of 7.5° (48 histograms).

Radial displacements, which are not presented in this article, can also be evaluated in a similar way [9].

The findings of Table 2 show that the thermally induced displacements on the Earth's surface can be as large as the interval $\langle -29.332; 40.070 \rangle$ m for tangential displacement over two years, an interesting finding that is not generally inconsistent with crustal motion. Some places on Earth move less and some move more, e.g., in areas of tectonic plate contact or faulting.

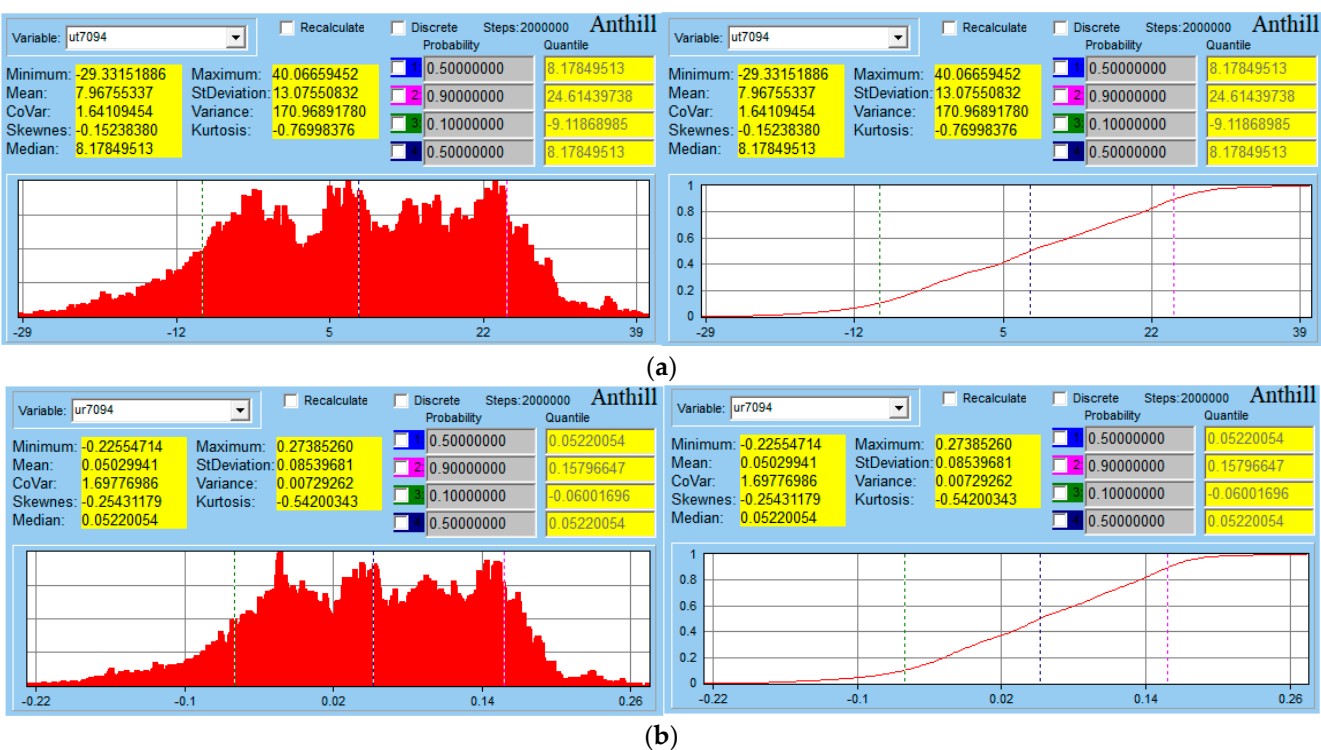

**Figure 14.** Histograms and distribution functions for node '7094' (**a**) tangential components of crustal surface displacement, (**b**) radial components of crustal surface displacement (sw MSC.Marc Mentat and Anthill).

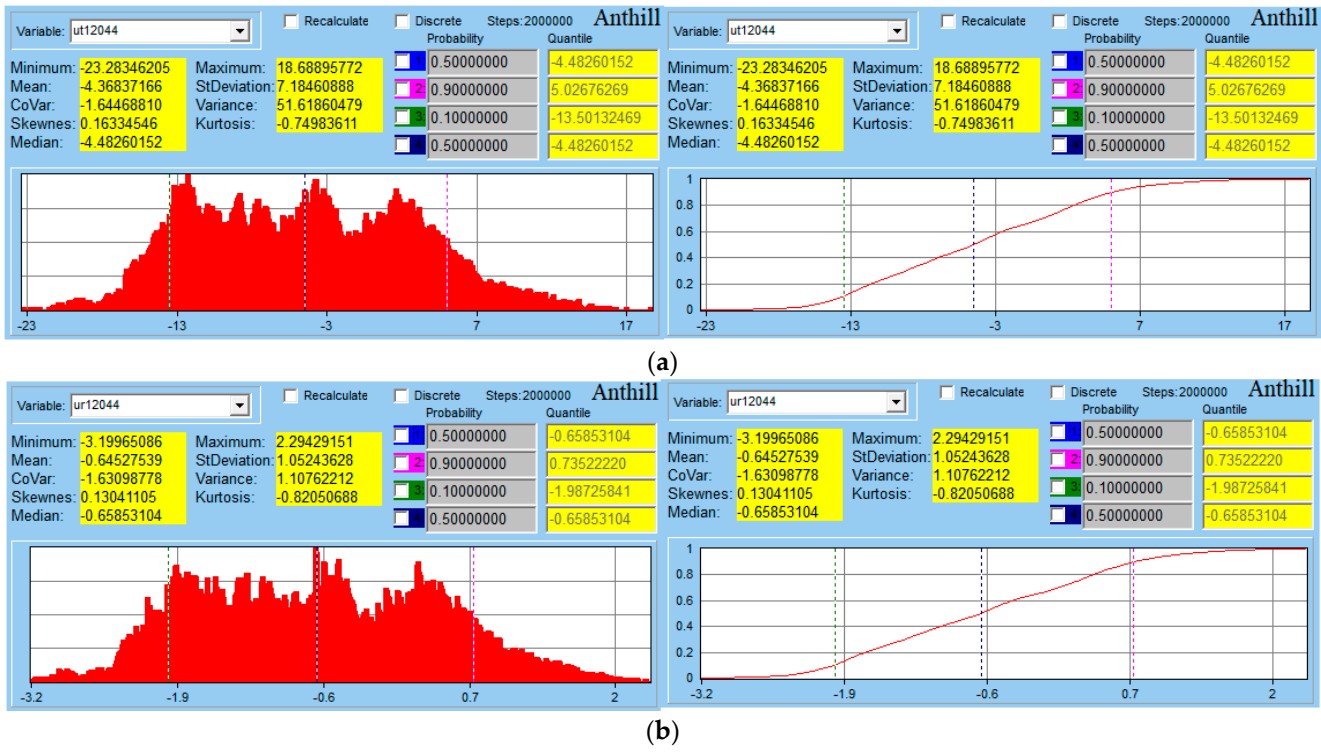

**Figure 15.** Histograms and distribution functions for node '12,044' (**a**) tangential components of crustal surface displacement, (**b**) radial components of crustal surface displacement (sw MSC.Marc Mentat and Anthill).

**Table 2.** Statistical parameters of tangential displacements.

| Node | Angle φ /deg/ | Min /m/ | Mean /m/ | Median /m/ | Max /m/ |
|------|---------------|---------|----------|------------|---------|
| 1 | 0.0 | −22.405145 | −4.016917 | −4.155867 | 19.706385 |
| 7646 | 7.5 | −13.167980 | 1.980264 | 2.203522 | 12.992962 |
| 7094 | 15.0 | −29.331777 | 7.971764 | 8.205548 | 40.070023 |
| 51 | 22.5 | −6.667464 | 1.772484 | 1.822645 | 8.890464 |
| 3 | 30.0 | −8.410809 | 0.015862 | 0.108814 | 8.880879 |
| 53 | 37.5 | −2.733061 | −0.369284 | −0.444654 | 2.724747 |
| 4 | 45.0 | −12.265832 | −2.029589 | −2.114735 | 11.494079 |
| 55 | 52.5 | −1.619803 | 0.019862 | 0.053834 | 1.584367 |
| 5 | 60.0 | −11.991985 | 2.225961 | 2.311249 | 13.202844 |
| 57 | 67.5 | −1.703642 | 0.148204 | 0.202752 | 1.321986 |
| 6 | 75.0 | −8.316151 | −1.156488 | −1.263755 | 9.357056 |
| 59 | 82.5 | −2.202902 | 0.168749 | 0.243539 | 1.710141 |
| 7 | 90.0 | −10.964751 | 1.914396 | 2.002252 | 11.641080 |
| 61 | 97.5 | −1.717808 | −0.088355 | −0.117302 | 1.903794 |
| 11491 | 105.0 | −14.292981 | −2.386571 | −2.480774 | 13.402903 |
| 63 | 112.5 | −2.136792 | −0.321966 | −0.361097 | 2.297582 |
| 11585 | 120.0 | −6.676715 | −0.051041 | −0.041540 | 6.316006 |
| 65 | 127.5 | −2.620005 | 0.196727 | 0.304310 | 1.918440 |
| 11782 | 135.0 | −10.374590 | 1.212906 | 1.450878 | 8.992166 |
| 67 | 142.5 | −3.431526 | 0.537606 | 0.601411 | 3.128304 |
| 11925 | 150.0 | −12.217613 | 2.033221 | 2.138069 | 12.652762 |
| 69 | 157.5 | −3.368985 | −0.578049 | −0.604307 | 3.254372 |
| 12044 | 165.0 | −23.284528 | −4.365703 | −4.469298 | 18.689545 |
| 71 | 172.5 | −2.412931 | −0.367422 | −0.407727 | 2.594586 |
| 13 | 180.0 | −10.843711 | 2.072278 | 2.139710 | 12.114409 |
| 73 | 187.5 | −2.344523 | 0.396470 | 0.427631 | 2.438536 |
| 12234 | 195.0 | −9.007515 | 0.961226 | 1.080069 | 7.542674 |
| 9225 | 202.5 | −2.045828 | 0.180401 | 0.245550 | 1.597627 |
| 12424 | 210.0 | −6.245059 | −0.379791 | −0.502522 | 7.471727 |
| 9372 | 217.5 | −3.101930 | −0.514486 | −0.545275 | 3.085332 |
| 12500 | 225.0 | −12.566509 | −2.093027 | −2.173286 | 11.838935 |
| 9471 | 232.5 | −2.061168 | 0.018279 | 0.060519 | 1.922392 |
| 12599 | 240.0 | −13.427477 | 2.207517 | 2.396578 | 13.803798 |
| 9594 | 247.5 | −2.078801 | 0.147412 | 0.237499 | 1.551424 |
| 12674 | 255.0 | −8.193241 | −1.127393 | −1.232754 | 8.960823 |
| 9736 | 262.5 | −3.634378 | 0.168002 | 0.216868 | 2.734815 |
| 19 | 270.0 | −18.844958 | 1.926130 | 1.995067 | 16.550423 |
| 9910 | 277.5 | −2.515986 | −0.117068 | −0.148072 | 2.376055 |
| 10004 | 285.0 | −14.114205 | −2.368588 | −2.561178 | 14.466366 |
| 12796 | 292.5 | −2.534931 | −0.399108 | −0.459958 | 2.579560 |
| 12919 | 300.0 | −6.537996 | −0.083045 | −0.093447 | 6.221886 |

**Table 2.** *Cont.*

| Node | Angle φ /deg/ | Min /m/ | Mean /m/ | Median /m/ | Max /m/ |
|---|---|---|---|---|---|
| 13017 | 307.5 | −2.130202 | 0.200543 | 0.266355 | 1.685076 |
| 13116 | 315.0 | −9.207633 | 1.157171 | 1.258941 | 8.282640 |
| 13259 | 322.5 | −3.271788 | 0.624200 | 0.663308 | 3.667678 |
| 13384 | 330.0 | −12.221221 | 2.082060 | 2.173141 | 12.821112 |
| 13569 | 337.5 | −2.657303 | −0.430821 | −0.459640 | 2.697734 |
| 10258 | 345.0 | −23.800367 | −4.473265 | −4.585868 | 18.998453 |
| 10135 | 352.5 | −8.195234 | −1.552937 | −1.599643 | 6.756131 |
| TOTAL | | −29.331777 | 0.063933 | 0.055363 | 40.070023 |

**Table 3.** Statistical parameters of radial displacements.

| Node | Angle φ /deg/ | Min /m/ | Mean /m/ | Median /m/ | Max /m/ |
|---|---|---|---|---|---|
| 1 | 0.0 | −0.247709 | 0.045662 | 0.048070 | 0.264486 |
| 7646 | 7.5 | −1.069984 | −0.220166 | −0.225239 | 0.744611 |
| 7094 | 15.0 | −0.225554 | 0.050253 | 0.052106 | 0.273853 |
| 51 | 22.5 | −2.949429 | −0.597693 | −0.609791 | 2.074209 |
| 3 | 30.0 | −2.840615 | −0.575473 | −0.589606 | 1.996382 |
| 53 | 37.5 | −2.451921 | −0.498885 | −0.509240 | 1.719541 |
| 4 | 45.0 | −2.948489 | −0.593957 | −0.604933 | 2.117883 |
| 55 | 52.5 | −3.122978 | −0.632979 | −0.646084 | 2.204687 |
| 5 | 60.0 | −3.688951 | −0.746602 | −0.762182 | 2.596254 |
| 57 | 67.5 | −3.284413 | −0.665639 | −0.679903 | 2.317962 |
| 6 | 75.0 | −3.600448 | −0.728494 | −0.744516 | 2.538881 |
| 59 | 82.5 | −2.910104 | −0.590158 | −0.600756 | 2.054802 |
| 7 | 90.0 | −3.126893 | −0.632771 | −0.646967 | 2.203372 |
| 61 | 97.5 | −3.260287 | −0.660954 | −0.672389 | 2.301363 |
| 11491 | 105.0 | −3.269007 | −0.658896 | −0.672805 | 2.344291 |
| 63 | 112.5 | −3.078366 | −0.623752 | −0.635934 | 2.171800 |
| 11585 | 120.0 | −3.613587 | −0.732255 | −0.748583 | 2.538011 |
| 65 | 127.5 | −3.173468 | −0.643556 | −0.659318 | 2.232398 |
| 11782 | 135.0 | −3.171930 | −0.642370 | −0.657984 | 2.232703 |
| 67 | 142.5 | −3.036209 | −0.616722 | −0.629598 | 2.138323 |
| 11925 | 150.0 | −3.896294 | −0.789001 | −0.806090 | 2.740027 |
| 69 | 157.5 | −3.991671 | −0.808510 | −0.825623 | 2.816905 |
| 12044 | 165.0 | −3.199840 | −0.645226 | −0.658717 | 2.294331 |
| 71 | 172.5 | −2.829153 | −0.573808 | −0.584828 | 1.996196 |
| 13 | 180.0 | −2.850070 | −0.577022 | −0.589402 | 2.007131 |
| 73 | 187.5 | −3.290911 | −0.667605 | −0.679138 | 2.323781 |
| 12234 | 195.0 | −3.451112 | −0.699088 | −0.712300 | 2.430043 |

**Table 3.** *Cont.*

| Node | Angle ϕ /deg/ | Min /m/ | Mean /m/ | Median /m/ | Max /m/ |
|------|------|------|------|------|------|
| 9225 | 202.5 | −3.446525 | −0.698456 | −0.712473 | 2.433441 |
| 12424 | 210.0 | −3.481797 | −0.705072 | −0.719768 | 2.452611 |
| 9372 | 217.5 | −3.153902 | −0.639660 | −0.651908 | 2.224569 |
| 12500 | 225.0 | −2.889573 | −0.583077 | −0.595777 | 2.063464 |
| 9471 | 232.5 | −3.225099 | −0.653850 | −0.670153 | 2.270460 |
| 12599 | 240.0 | −3.643746 | −0.739617 | −0.755094 | 2.553612 |
| 9594 | 247.5 | −3.900572 | −0.792134 | −0.808253 | 2.746832 |
| 12674 | 255.0 | −3.421864 | −0.694008 | −0.705438 | 2.417951 |
| 9736 | 262.5 | −2.969488 | −0.607632 | −0.618616 | 2.114656 |
| 19 | 270.0 | −3.264306 | −0.662024 | −0.676559 | 2.271688 |
| 9910 | 277.5 | −3.891178 | −0.791065 | −0.807631 | 2.689200 |
| 10004 | 285.0 | −3.323049 | −0.665437 | −0.681295 | 2.371287 |
| 12796 | 292.5 | −3.731120 | −0.756405 | −0.773357 | 2.623065 |
| 12919 | 300.0 | −3.443254 | −0.697380 | −0.711933 | 2.418230 |
| 13017 | 307.5 | −3.195928 | −0.648019 | −0.661250 | 2.255990 |
| 13116 | 315.0 | −3.107899 | −0.628968 | 2.190442 | 2.190442 |
| 13259 | 322.5 | −3.210917 | −0.650901 | −0.664147 | 2.267816 |
| 13384 | 330.0 | −3.473598 | −0.702698 | −0.716477 | 2.446307 |
| 13569 | 337.5 | −3.841410 | −0.777634 | −0.794236 | 2.709838 |
| 10258 | 345.0 | −1.664193 | −0.333632 | −0.341247 | 1.236957 |
| 10135 | 352.5 | −1.383713 | −0.281419 | −0.288036 | 0.969561 |
| TOTAL | | −3.991671 | −0.613224 | −0.565425 | 2.816905 |

The findings in Table 3 show that the displacement at the Earth's surface can be as large as $\langle -3.992; 2.817 \rangle$ m for radial displacement over two years. This is interesting and, again, is not inconsistent with crustal motion.

The above-mentioned displacements are usually quasi-static but, in extreme situations associated with earthquakes, they can also occur in a dynamic manner.

The histogram of the resulting radial and tangential displacements (stochastic evaluation of tangential and radial displacement) is shown in Figure 17.

A large number of different relationships can be used to estimate the error of the Mote Carlo method. For example, consistent with the literature [45,46], the error of the Monte Carlo method can be approximately estimated by the relation:

$$\text{error}_{\text{MonteCarlo}} = \frac{\text{StDeviation}}{\sqrt{\text{Steps}}}, \tag{3}$$

where 'StDeviation' is the standard deviation and 'Steps' is number of Monte Carlo simulations.

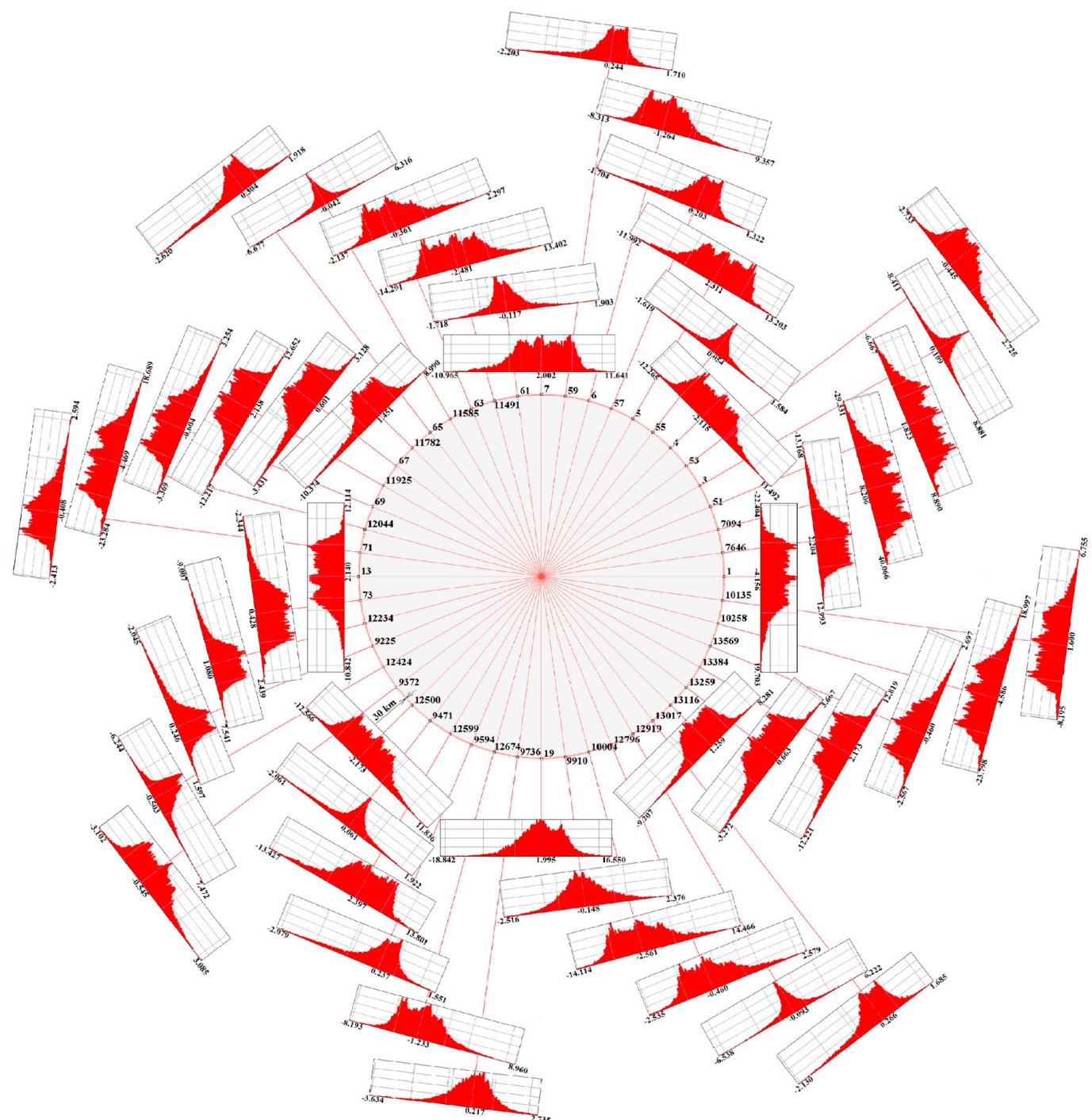

**Figure 16.** Histograms of the tangential component of the displacement at nodes on the Earth's outer surface induced by thermal radiation from the Sun.

From the data in Figure 17a, Equation (3) gives $\frac{3.90418181}{\sqrt{2000000}} = 0.00276$ and, from the data in Figure 17b, Equation (3) gives $\frac{1.05741154}{\sqrt{2000000}} = 0.00075$. The error is too small, and the stochastic approach is reliable.

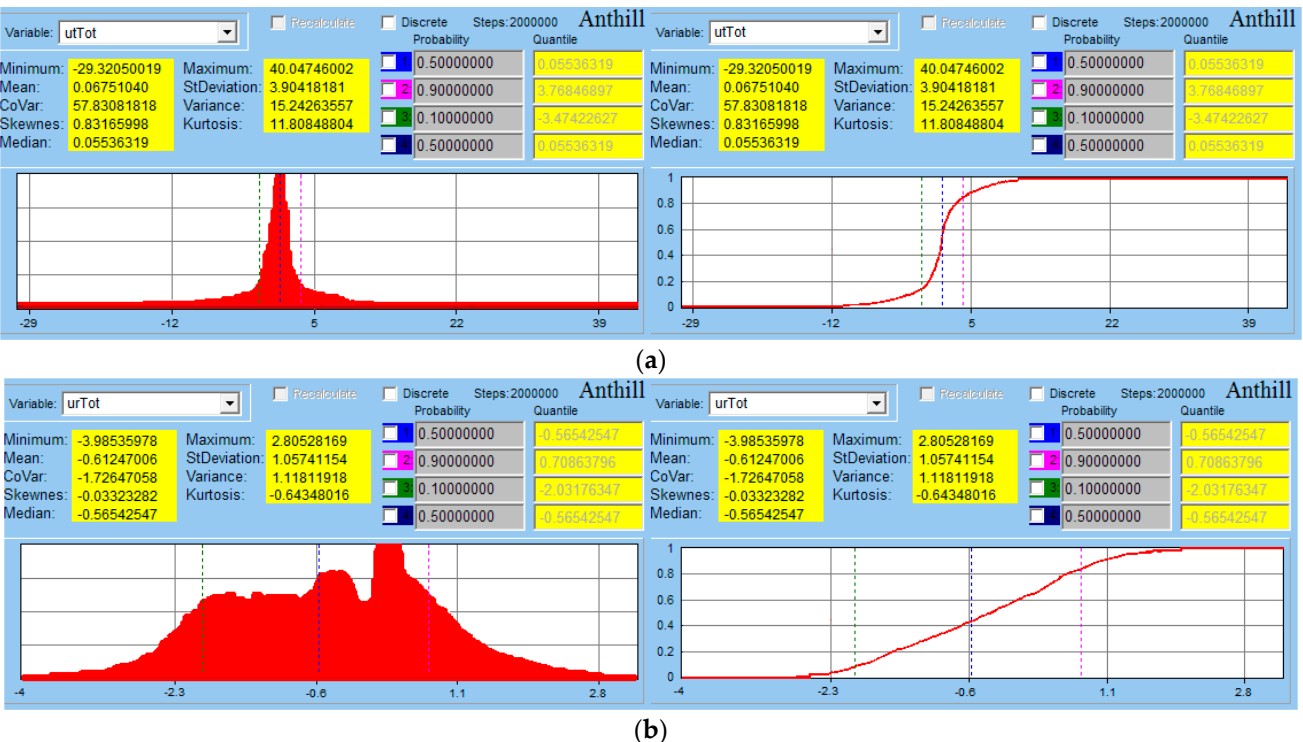

**Figure 17.** Histograms and distribution functions of (**a**) tangential component of total crustal surface displacement, (**b**) radial component of total crustal surface displacement (sw MSC.Marc Mentat and Anthill).

## 4. Discussion

An initial new model of the Earth's heat loading has been developed. Its advantage is the simplicity of the plane strain problem and the original combination of the finite element method and the probabilistic Monte Carlo Method. FEA is performed for the non-stationary and non-linear coupled geomechanical problem (i.e., temperature task and structural task).

A simple measure of the material Inhomogeneity of the Earth's crust (partitioned into 24 materials) was respected in a simple way.

The interaction between the crust and the upper mantle is replaced by the bilateral Winkler elastic foundation boundary condition.

This results in a simplified model, which respects many random (pseudo-random) inputs, i.e., $2 \times 10^6$ Monte Carlo simulations. Nonlinear geomechanics tasks, in the case of large deformation caused by Sun heating, were solved.

From the stochastic modelling, the tangential and radial displacements of the Earth's surface and crustal stresses are obtained, and their values can initiate the possible fractures of the Earth's crust connected with its tectonics.

The model is considered to be continuous and, therefore, does not allow for the formation of cracks in the Earth's crust. However, the obtained stresses and displacements clearly confirm sufficient capacity to induce tectonic changes in the Earth's surface. The periodic and inhomogeneous thermal radiation from the Sun acting on the Earth's surface is sufficient to induce these tectonic changes. This is the important finding for understanding the complexity of the Earth's crust behavior.

It can be said that tectonic changes in the Earth's crust are also caused by thermal radiation from the Sun (external heating of the Earth), which is still a new idea. Our results are not inconsistent with earlier foreign works [26,28,29] or our works [9,15,18,19,30,31].

As is well known, tectonic changes in the Earth's crust are also due to tidal effects from the gravitational pull of the Sun and Moon, centrifugal forces from the Earth's rotation,

and internal magmatic processes within the Earth. The influence of tidal, centrifugal, and internal magmatic effects is not addressed in this paper.

In the future, the dynamic and cyclic loading of the Earth's crust from tidal effects and magmatic processes can also be considered, or a more complex spatial (3D) model can be developed that could also account for the other geological and geomechanical phenomena mentioned above. In the future, by applying mechanical contacts between the Earth's plates, it will be possible to deal with tectonic phenomena on our own or foreign planets in a similar but more complex and complicated way.

The important influence of thermal radiation on the Earth's tectonics are new findings, proved by our work and by the different models, but consistent with the literature [9,15,18,19,26,28–31].

## 5. Conclusions

Using a simple model for the heating of the Earth from a heat source, i.e., the Sun, over a two-year period, the displacement fields and equivalent mechanical stresses (von Mises) within the Earth's crust were obtained.

The Sun causes relatively large displacements in the Earth's crust during annual periods (and, to a lesser extent, over a daily period), reaching up to 40.07 m in the tangential direction and 3.99 m in the radial direction at the crustal surface. However, it is a slow process that is almost imperceptible to humans, compared to earthquakes, where the crustal displacement is visible to the naked eye.

The numerical analysis further revealed stresses that reached up to 52 MPa. These stresses, from temperature alone, are normal for the Earth and are not very damaging to it but can cause tectonic movements. However, in combination with other geomechanical phenomena, e.g., tidal effects, the Earth's rotation, or convection currents within the Earth, etc., it can cause faults in the Earth's crust.

Our findings, with respect to the important influence of solar thermal heating on the Earth's tectonics, are new as well as proven by different models, but consistent with the chosen references. We have proved that the Sun can 'move' the Earth's crust.

**Author Contributions:** Methodology, K.F.; Formal analysis, I.W.; Writing – original draft, D.Č., K.F. and I.W. All authors have read and agreed to the published version of the manuscript.

**Funding:** This article was supported by Specific Reasearch SP2023/027 "Application of Modern Computational and Experimental Approaches in Applied Mechanics"; Specific Research SP2022/26 "Computational and experimental modeling in the tasks of applied mechanics and biomechanics" and by international projects CZ.02.1.01/0.0/0.0/17_049/0008441 "Innovative Therapeutic Methods of Musculoskeletal System in Accident Surgery" and CZ.02.1.01/0.0/0.0/17_049/0008407 "Innovative and additive manufacturing technology—new technological solutions for 3D printing of metals and composite materials" within the Operational Programme Research, Development and Education financed by the European Union and from the state budget of the Czech Republic.

**Institutional Review Board Statement:** Not applicable.

**Informed Consent Statement:** Not applicable.

**Data Availability Statement:** Not applicable.

**Conflicts of Interest:** The authors declare no conflict of interest.

## Nomenclature

| Name | Physical Unit | Explanation |
| --- | --- | --- |
| ALA | | Monitoring system for measuring |
| c | $Jkg^{-1}K^{-1}$ | Specific heat of Earth's crust |
| E | Pa | Young's modulus of Earth's crust |
| Earthcylinder | | Simplified model of geoid of Earth |
| error$_{MonteCarlo}$ | 1 | Possible error of Monte Carlo Method |

| Name | Physical Unit | Explanation |
| --- | --- | --- |
| f | | Function |
| FE | | Finite element |
| FEA | | Finite Element Analysis |
| FEM | | Finite Element Method |
| K | $Nm^{-3}$ | Modulus of the foundation |
| material1, material2, ... material24 | | Name of materials in Earth's crust |
| Max | | Maximum value |
| Mean | | Mean value |
| Median | | Median value |
| Min | | Minimum |
| MSC.Marc Mentat | | Finite element software |
| Node | | Node of finite element mesh |
| R | m | Radius of Earth's crust |
| $R_1$ | m | Inner radius of Earth's crust |
| $R_2$ | m | Outer radius of Earth's crust |
| StDeviation | | Standard deviation |
| Steps | | Number of Monte Carlo simulations |
| Stress HMH | | Equivalent von Mises stress |
| T | K | Temperature |
| $T_1, T_2, ... T_{24}$ | K | Temperature loading on section 1, 2, ... , 24 of Earth's crust, see Figure 3 |
| $T_c$ | K | Initial temperature on inner radius of Earth's crust, see Figure 5 |
| $T_{initial}$ | K | Initial temperature of Earth's crust from interval $\langle T_c; T_P \rangle$ |
| $T_P$ | K | Temperature on inner radius of Earth's crust |
| t | s | Time |
| $u$ | m | Total displacement in point of Earth's crust |
| ut1 | m | Tangential component of crustal surface displacement for node 1 |
| ut7094 | m | Tangential component of crustal surface displacement for node 7094 |
| ut12044 | m | Tangential component of crustal surface displacement for node 12044 |
| utTot | m | Tangential component of total crustal surface displacement |
| ur1 | m | Radial component of crustal surface displacement for node 1 |
| ur7094 | m | Radial component of crustal surface displacement for node 7094 |
| ur12044 | m | Radial component of crustal surface displacement for node 12044 |
| urTot | m | Radial component of total crustal surface displacement |
| $u_x$ | m | Displacement in X axis direction |
| $u_y$ | m | Displacement in Y axis direction |
| X | m | Axis of coordinate system |
| Y | m | Axis of coordinate system |
| Z | m | Axis of coordinate system |
| α | $K^{-1}$ | Thermal expansion coefficient of Earth's crust |
| φ | deg | Angle of Earth's crust |
| λ | $Wm^{-1}K^{-1}$ | Conductivity of Earth's crust |
| μ | 1 | Poisson number of Earth's crust |
| ρ | $Kgm^{-3}$ | Density of Earth's crust |
| $\sigma_{HMH}$ | Pa | Equivalent von Mises Stress |
| $\sigma_1, \sigma_2, \sigma_3$ | Pa | Principal stress |

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
