# Peer review of "Analysis of the Influence of Thermal Loading on the Behaviour of the Earth’s Crust"

_applsci, doi:10.3390/app13074367_

Round 1

Reviewer 1 Report

This model definitely cannot be representative for the Earth crust because any type of heterogeneity has been considered. Lithospheric thicknesses includes many elements in which their compositions provide thousands of minerals and these minerals then are considered the elements of many different rock types and formation units. Therefore, in the simplest form you must and have to consider three stage of heat conductivity and stress transferring in which this model never ever can be representative for all of them. Referring to Figure 1 and 2 the considered crust is very simplified condition in which can be evaluated with shell analysis. In a heterogeneous media you are not allowed to consider the similar portions for different minerals and thus materials. Furthermore, the thickness of crust is totally different in continental and sea. Considering other aspects, the given model is just a simple circle with constant thickness and equal portion of material that is totally far from the reality. Where did you consider any uncertainty for the used data? What are the used input parameters? How the uncertainty for the output has been evaluated and treated?. https://link.springer.com/article/10.1007/s11053-022-10051-w#citeas will give you deep insight.  The characteristics and physic-mechanical properties of surface outcrops of the crust with the similar material beneath it (e.g. E, UCS, poison ratio, heat conductivity, … ) due to weathering, alteration and many other land surfaces in a same area obviously are not constant while you took all of them constant. This is totally in scientific border is rejected.

The used model just analyzes the nonlinearity, not the actual structure and thus making different modeling assumptions will get different results for the same structure. Moreover, the response of a structure is sensitive to the strengths and stiffnesses of its components, and the actual properties are not known accurately. Capacity design can greatly reduce the uncertainty. No information regarding these problems and treatment methods has been given.

Another wondering point is the claimed statement in Line 15 ‘over a time span of 2 years’???? You measured it for two years???? If yes, without any doubt must be clarified how, with which tools, from which depth, for how many points, at where (locations), for which time interval, in how many countries, and … what about the spatially distribution of collected station points???? and ….

Can this model provide not only correct inferences but also some explanation for the underlying phenomena? Can users gain a mechanistic understanding of this model? Definitely NOT.

You have used time series data in which their analysis cannot provide any generalization from a single study and is difficult in obtaining appropriate measures. It also suffers from accurately identifying the correct model to represent the data. Highlight how these problems have been overcome?

The past data of the time-series is not enough to predict the future. Multiple additional features should be taken into account to get good forecasts. How did you judge the efficiently dealing with outliers? How to efficiently deal with multiple periodicities? Your works severely suffers from responses to these questions.

Some other comments:

-          Each abbreviation in the first use must be defined in full format, even it is widely recognized.

-          The English of the work suffers from significant and frequent linguistic flaws, word repetitions, long and vague statements as well as punctuation issues. Simply can be seen from the first sentence in Abstract. Must and have to be revised by native expert or verified proofread systems.

-          Abstract should be short, concise but self-informative and to the point. Here is just lecturing and there is no chance to get what the novelty is. How many samples, from which date interval, what are the dependent and independent variables? Spatially distribution of samples … Overall, this part strongly is rejected.

-          Keywords should be representative and available in both Abstract and context. Almost all the used words are general terms. FEM is a keyword? Solar thermal??? And …

-          The last paragraph of the Introduction should be assigned to brief summary of applied method and bolded findings.

-          Without any doubt null, dull and far too short introduction in terms of both technically and literature review. Must and have to be updated with technical literature reviews. From the whole introduction, there is no chance to get quick catch on what the problem is and what goal you are pursuing. What are the main gaps of previous applied models and which gap of them based on which method is going to be covered? What is the applied method and its advantage rather than previous ones? What motives for? The main significant of contribution?

-          What about on covering the lack of ability to be spatially invariant to the input data?

-          Did you code the procedure? If yes, with what programing language? How the weights are saved? With what command it can be recalled? When you have different involved parameters definitely sensitivity analyses for model calibration through the weight database must be carried out to show their influence on the results. Given results only uncovers the samples and won't determine what variables have the most influence. Extraneous variables might interfere with the information and thus outcomes can be adversely impacted by the quality of the work. Look at https://www.sciencedirect.com/topics/medicine-and-dentistry/sensitivity-analysis, https://iwaponline.com/jh/article/22/3/562/72506/Updating-the-neural-network-sediment-load-models and ...

-           How did you reduce the variability of nonlinear analysis?

-          Each data type has some certain value of maximum and minimum, and if the data stored under certain data type crosses these permissible values, then overflow and underflow occurs. This leads to truncation and round off errors in which your FEM-based model always incorporate with. So, how did you evaluate these errors and with what criteria?

-          What will happen in different material distributions? Something like as discussed in https://www.mdpi.com/2220-9964/10/5/341/htm and ...  

-          What are the used assumptions? Type of mesh and why? How many meshes? How the boundary condition has been defined? Where are the used data for model construction? How they have been incorporated?

-          Where is the evidence to show the accuracy performance? Comparing to previous models, did you use it exactly as it was or modify it for your case? Given information? What about the reality of validation and verification criteria.

-          FEM needs large amount of data as input for the mesh used in terms of nodal connectivity and other parameters depending on the problem in which the output result will vary considerably. How this problem has been treated???

-          Given a differential equation to be solved, one must find the corresponding functional. This is an extremely difficult thing to do in general. How this drawback has been solved??

-          Idealization of real-life objects can't be exact for complex shapes. FEM yields approximate solution. It is tried to minimize the error over the whole domain, as a result we get exact solution at nodes only.

-          Where are the reliability-based and uncertainty analyses?

-          According to above mentioned comments this paper doesn't have real discussion

To save the time, I gave the rest up.

Author Response

Dear reviewer,

first of all, we would like to thank you for the time you took to write the review. We will try to respond to your comments. However, it is not possible to keep the response brief as you have covered a wide range of issues from many disciplines in the review. We will try to make it clear that we have dealt with all of this in our search for a solution.

RNDr. Ivo Wandrol, Ph.D.

Reviewer 2 Report

The figures need to be improved. Figures 10, 11, and 13 in particular should be presented using output data rather than a snapshot taken by the software

Author Response

Dear reviewer no. 2

First of all, we would like to thank you for the time you took to write the review. We will try to respond to your comments. However, it is not possible to keep the response brief as you have covered a wide range of issues from many disciplines in the review. We will try to make it clear that we have dealt with all of this in our search for a solution.

RNDr. Ivo Wandrol, Ph.D.

Reviewer 3 Report

applsci-1847846   Review. 

This paper is a numerical study of the effect of a varying surface temperature on the Earth’s crust.  A simplified model is used, meaning that the results may be more qualitative than physically and realistically correct.  The paper could be ultimately published after some changes.  Some comments and questions follow. 

[1] Including a 3D plot of T=f(φ,t) might be helpful and informative. 

[2] Eq. (1): Why just use σ_HMH?  Why not compute all the components of the stress tensor? 

[3] Figure 5 and 7:  It seems strange that the stress and displacement should oscillate so much, with a main period of about 200 days.  Can the authors explain this? 

[4] Figures 9-11: What do the numbers on the horizontal axes indicate? 

[5] Table 2 and 3: It seems strange that the tangential displacements can vary so much, from -30 m to +40 m with a mean of about 8 m, for node 7094 for example.  These amounts seem excessive.  The radial displacements also seem excessive.   This would affect GPS and other locations systems, correct?  The authors state that these findings are “ generally not inconsistent with crustal motion”.  Can some references be given for this statement?  Are such motions observed or measured for the crust?  Can references be given for that?  Or are these motions exaggerated because of the cylinder model?  If the spherical nature of the Earth could be implemented in the calculations, would the results be significantly different?

[6] Because of the simplifications made, i.e., modeling the crust as piecewise isotropic, ignoring gravitational, tidal and centrifugal effects and interior magmatic processes, and using a planar cylinder model, it is difficult to say how accurate and realistic the findings are.  As I indicated in the previous comment, I am having difficulty believing them.  Can the authors argue that the results are physically realistic?  Or are the results are mostly qualitative? 

[7] Line 317: It is also hard to believe that the thermal radiation from the sun can cause significant tectonic changes.  Is there evidence for this? 

Author Response

Dear reviewer no. 3

First of all, we would like to thank you for the time you took to write the review. We will try to respond to your comments. However, it is not possible to keep the response brief as you have covered a wide range of issues from many disciplines in the review. We will try to make it clear that we have dealt with all of this in our search for a solution.

RNDr. Ivo Wandro Ph.D.

Reviewer 4 Report

In this paper, the strain-stress analysis of the Earth's crust under external loading is focused, specifically the influence of cyclic changes in the surface temperature field on the stress and strain inside the crust over a time span of 2 years is investigated. This manuscript is worth of publication after some minor modifications:

(1): Line 28-33: The Earth's crust is affected by many external influences. However, this paper focuses only on the effect of cyclic surface temperature changes on the stress and strain of the Earth's crust. Why?

(2): Line 38-39: “Due to the complexity of the whole process, the Finite Element Method (FEM) was used and the problem was solved as a planar one.” The necessary references should be added to justify this simplification.

(3): It is suggested in the introduction to state the innovations of this manuscript.

(4): Line 64-66: “The Earth's crust is composed of lithospheric plates. The crust is mainly made up of rocks and minerals, where various faults, fractures and other geological formations occur. In addition to rocks and minerals, there are also gases and water. It is therefore inherently a highly anisotropic, heterogeneous material.” Necessary literature should be added to support this statement, and the following references are recommended:

Shen JH, Wang X, Cui J, Wang XZ, Zhu CQ. Study on the shear characteristics of calcareous gravelly sand considering particle breakage. Bulletin of Engineering Geology and the Environment.

(5): Line 189-190: “From Fig. 4, the stress attenuation towards depth is evident. Similar was the case for the other sections.” These are only simulation results, and the authors should elaborate on the reasons for such results.

(6): In order to improve the readability of the manuscript, it needs an exhaustive revision by a native English speaker.

Author Response

Dear reviewer no. 4

First of all, we would like to thank you for the time you took to write the review. We will try to respond to your comments. However, it is not possible to keep the response brief as you have covered a wide range of issues from many disciplines in the review. We will try to make it clear that we have dealt with all of this in our search for a solution.

RNDr. Ivo Wandrol, Ph.D.

Round 2

Reviewer 1 Report

Dear authors

I read your response carefully. By the given clarification at the preface of responses, I may agree with some of your responses but unfortunately not for all. You can use the line number embedded in template of MDPI as very good identifier to dedicate the line numbers corresponding to each comment where you have modified. Finally, we are here to help each other and increase the quality of the work, so, considering the following concerns is important:

1. I couldn’t’ understand ‘Un our model’!!! Just for example, concerning the uncertainty, you got misunderstood on the for example suggested reference. It is not the case for citing in the context, the core of this comment is to remind you how the uncertainty and inclusion of impurities for clean datasets have been taken into account. I just remind you First of all, the suggested reference presents a developed state-of-the-art approach for uncertainty quantification. Second, in terms of computational analysis you can find the concept of used formula, development, and relations. Third, it has practically and computationally has been compared with MC and other methods. Fourth, it is not limited in Sweden but just has been applied for data from Sweden, same as all other developed techniques to show the practical capability. Just as simple facial interpretation, you have used MC and FEM so can I say that this type of application is just limited to simple FEM??

2. ‘today it is not possible to make geomechanical model of a real whole Earth’!!! I agree but for simulation you have used FEM with defined geomechanical characteristics close to real conditions for the defined elements which will further be distributed along the model.

3. I didn’t get feedback for this comment. Who is the host of this website and how the gathered data here are verified? Could you please clarify it?

4. ‘It is written there.’!!! Couldn’t follow you.

5. I asked for how the model can identify the accurate time dependent model with the least divergency?.

6. You have a FEM model and the output. so, based on the used data is somewhat predicting. Referring to #5, the time series can be used for future data and therefore covers this comment. Anyway, I would like to know how you removed the outliers as always you will have anomalies.

13. Spatial invariance simply means that shifting the input signal results in an equally shifted output signal. I think looking at https://dahleh.lids.mit.edu/wp-content/uploads/2011/09/bamieh01.pdf, https://ieeexplore.ieee.org/document/6957400 or ... Can be beneficial.

14. In the reference list the given software must be properly cited. Date, producer, … and other identifiers which is required for a reference based on template of Applied Sciences.

15. The comments #14 and #15 are in the same direction. ‘Basically, we were interested in how the nonlinearity at the interface of the sections would manifest’, you didn’t get my point. Mathematically, when you have a function with different variables, in many cases with a what if analysis the leas effective factors are removed. This will help for producing more calibrated model with the similar results that not only the least important variables have been removed but also the help to increase the process speed. This is very popular and known concept in modeling problems.

16. The forum like https://stackoverflow.com/questions/62935799/generally-how-do-i-prevent-integer-overflow-from-happening-in-c-language, https://www.wasyresearch.com/common-problems-in-numerical-computation-from-data-overflow-rounding-error-poor-conditioning-to-memory-leak/ can help you for wider view in this comment.

17. Agree that the suggested work is within regional scale, the core which I emphasized on was the effect of distributed material and their topography (location) within your model. Hope that you get my point.

19. Dear colleagues, accuracy performance means the used metric to show how we can rely to the results. The random simulation of MC is not the response but plays a role for that.

24. When we talk about  discussion, it means for using for example validation data, verification metrics, comparing with other models, … discussion on limitations of presented method!! Can you prove the convergence or stability of the proposed method?? Any explicit discussion to illustrate the limitations, pitfalls and practical difficulties of applied models under certainty?? the work lacks for a prior impact assessment and cost-benefit analysis and did not anticipate solutions for the possible consequences in advance.

About the references:

In some cases, there are several inconsistencies, for example:

1. Wikimedia cannot be a scientific references. Suggested to find validate resources.

2. should be modified. ‘Geothermal Energy’ is the book title. So, You must put ‘Basic concept’ as the title of chapter of the book.

5. Should be similar to others and supplemented by full address https:….

7. Each journal has ISSN, so there is no require to use ISSN.

8. Follow the instruction for authors in the case of dissertation.

10, 11. Basically are similar.

13, 14, 15. Follow the instruction for authors in giving the number of pages.

16, 20, 21, 22, 24, 25, 30. should be supplement by doi web link.

23. the name of the first author should be modified as ‘Abbaszadeh Shahri, A.’.

31. follow the instruction for authors in proceedings.

37, 39. Recheck it. Follow the instruction for authors.

41. Place and publisher.

42. date, used version, … follow the instruction for authors.

44. Date, used software, place, …

45. I couldn’t follow that.

Author Response

Dear reviewer no. 1

    Thank you for your work. We checked, corrected and extended our article based on other reviews.

    At the beginning, there was 4 reviewers. We passed through (i.e. satisfied) 3 of them. You are the last one which is remaining. However, we correct our article again.

   Our answers are written in red. Any changes are in green colour in file "Article-changes-in-green.docx".

With regards Karel Frydrýšek and its team.

Reviewer 2 Report

Figures should be reproduced based on the output data, not photos from software

Author Response

Dear Reviewer No.2
The answer to your question/request is in the attached Word document.

Sincerely
the team of authors
